# Empirical Assessment Tool for Bathymetry, Flow Velocity and Salinity in Estuaries Based on Tidal Amplitude and Remotely-Sensed Imagery

**Jasper R. F. W. Leuven** * , **Steye L. Verhoeve, Wout M. van Dijk** , **Sanja Selaković** and **Maarten G. Kleinhans**

Faculty of Geosciences, Utrecht University, Princetonlaan 8A, 3584 CB Utrecht, The Netherlands; s.l.verhoeve@students.uu.nl (S.L.V.); W.M.vanDijk@uu.nl (W.M.v.D.); S.Selakovic@uu.nl (S.S.); m.g.kleinhans@uu.nl (M.G.K.)

\* Correspondence: j.r.f.w.leuven@uu.nl

**Abstract:** Hydromorphological data for many estuaries worldwide is scarce and usually limited to offshore tidal amplitude and remotely-sensed imagery. In many projects, information about morphology and intertidal area is needed to assess the effects of human interventions and rising sea-level on the natural depth distribution and on changing habitats. Habitat area depends on the spatial pattern of intertidal area, inundation time, peak flow velocities and salinity. While numerical models can reproduce these spatial patterns fairly well, their data need and computational costs are high and for each case a new model must be developed. Here, we present a Python tool that includes a comprehensive set of relations that predicts the hydrodynamics, bed elevation and the patterns of channels and bars in mere seconds. Predictions are based on a combination of empirical relations derived from natural estuaries, including a novel predictor for cross-sectional depth distributions, which is dependent on the along-channel width profile. Flow velocity, an important habitat characteristic, is calculated with a new correlation between depth below high water level and peak tidal flow velocity, which was based on spatial numerical modelling. Salinity is calculated from estuarine geometry and flow conditions. The tool only requires an along-channel width profile and tidal amplitude, making it useful for quick assessments, for example of potential habitat in ecology, when only remotely-sensed imagery is available.

**Keywords:** estuary; morphology; rapid assessment; bathymetry; flow velocity; salinity; tool; remotely-sensed imagery

## 1. Introduction

Estuaries are characterised by fresh water inflow at the landward boundary and an open connection to the sea. Within these boundaries, tidal flows form dynamic patterns of channels and bars (Figure 1a). Many of these systems are managed to balance the needs of flood safety, access to harbours and ecological quality. The main channels in large systems such as the Western Scheldt, Elbe and Yangtze are dredged for access to harbours [1], while intertidal area on bars and estuarine shorelines forms valuable ecological habitat [2]. Depth, inundation time, flow velocity and salinity are the prime abiotic factors that affect living organisms in estuarine environments [2–4]. However, the general lack of information on these abiotic factors limits the prediction and sustainable management of habitat area in most estuaries [5]. Additionally, the effects of human interventions and rising sea-level on the hydrodynamic conditions, equilibrium morphology and, therefore, also on ecology are largely unknown. Unfortunately, data to study these effects are generally only available for a few economically

exploited estuaries that are already under great pressure by human influence. Data are lacking for most other systems around the world. Here, we explore to what degree it is possible to globally estimate estuarine characteristics, such as bed elevation, inundation duration, flow velocity, and salinity based on limited but widely publicly available data: remotely-sensed imagery and tidal range at the estuary mouth. To aid application and further investigation, we make the toolbox available.

Currently, several alternative methods are available to study the equilibrium morphology and hydrodynamic conditions of estuaries: in situ measurements, numerical modelling [6–8], idealised form models [9], stability analyses [10–12] and stability relations [13–16]. Only numerical models, for example Delft3D, can predict the spatial patterns of bed elevation, flow velocity and salinity with considerable accuracy at the spatial detail that is required for the purpose of habitat estimation. However, morphological models are computationally expensive and require calibration and detailed measurement of boundary and initial conditions. In situ measurements are an alternative, but they are expensive and equally time-consuming. For example, Rijkswaterstaat typically covers 2 km$^2$/day in the Western Scheldt. While linear stability analysis may predict typical estuarine bar patterns, it is not straightforwardly applicable to fully developed bars and estuaries with irregular planforms.

For converging ideal estuaries, predictive relations are already available between width-averaged bed elevation and hydrodynamic conditions along the system [9,17,18] because of the imposed constraint that the tidal damping by friction is balanced by the gain in energy by convergence. While these concepts are useful when applied to end-members, they leave the more common estuaries with irregular planforms unexplored. We recently found that the concept of an ideal estuary can be used to predict bed elevation and bar patterns in estuaries with irregular planform shapes as a function of the deviation from the ideal shape [19,20]. However, we still lack empirical or theoretical relations that describe flow velocity in these non-ideal systems, while this is an essential input variable for salinity predictions [21–23] and habitat suitability [24,25]. Based on observations in the Western Scheldt, we hypothesise that a relation exists between the depth and tidal flow velocity amplitude (Figure 1a,b). Here, we derive a predictor for flow velocity as a function of bed elevation to complement the available relations for bed elevations, bar patterns and salinity in estuaries with irregular planforms.

Here, existing empirical relations and theoretical concepts were selected and combined in a fast-to-apply tool that predicts equilibrium morphology and hydrodynamic conditions. To assess the applicability, we test it against a range of estuaries and other end-member systems, characterised by varying degrees of tidal and river influence [9,26] as well as varying degrees of adaptation to boundary conditions [9,27] and varying degrees of bar presence (Figure 2). These systems are a river-dominated delta branch [26], a tide-dominated delta branch (an ideal estuary cf. Savenije [18]), an alluvial estuary filled with tidal bars [19,26], a relatively wide estuarine valley partially filled with bars [26,28], and a tidal basin lacking fluvial input [29–31] (Figure 2). In the discussion, we show applications for management of estuaries, palaeogeographic reconstructions and quantifying habitat area based on species-specific preferences.

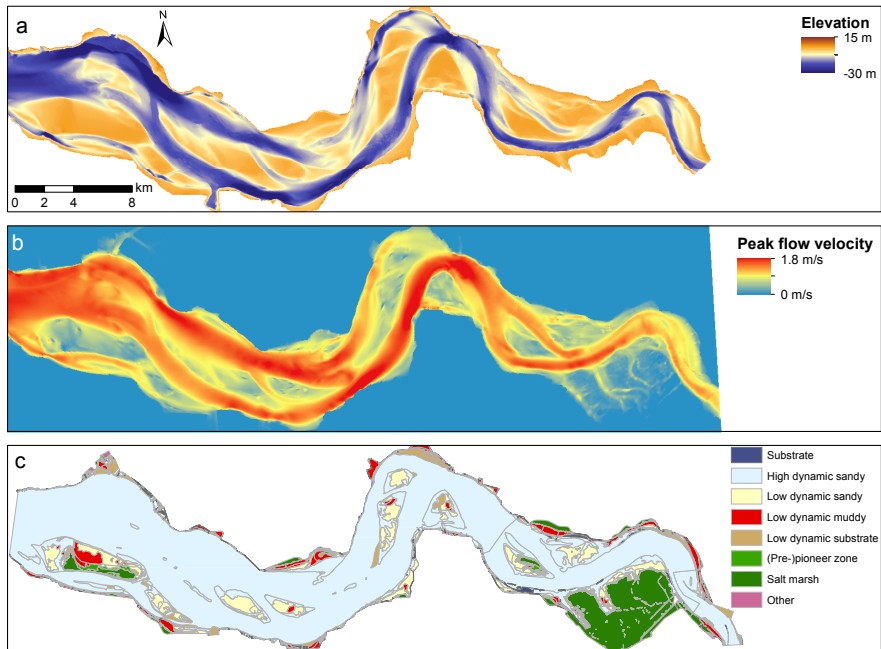

**Figure 1.** (**a**) bathymetry of the Western Scheldt in 2012 from the mouth (left) to the Dutch-Belgian border (right); (**b**) peak tidal flow velocities from a calibrated hydrodynamic model of Rijkswaterstaat (SCALWEST, [32]) for the bathymetry of the Western Scheldt in 2009 and (**c**) ecotope map of the Western Scheldt (2012). Bathymetry, hydrodynamic output and ecotope map were obtained from Rijkswaterstaat.

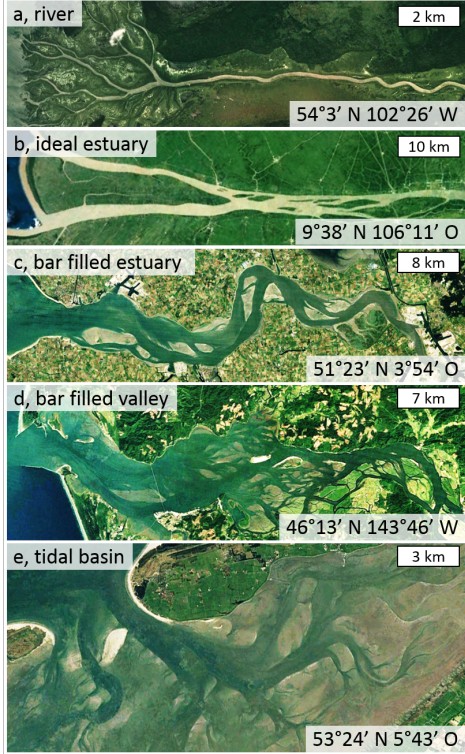

**Figure 2.** Examples of the systems used in this study. (**a**) a delta branch of a distributary of the Saskatchewan River (Canada); (**b**) the Tran De branch of the Mekong Delta (Vietnam); (**c**) the Western Scheldt estuary (The Netherlands); (**d**) the Columbia River estuary (USA); (**e**) the Ameland inlet and tidal basin (The Netherlands).

## 2. Methods

Below, first, the general approach and minimum data required to run the tool are described. Then, the general structure and functions used in the tool are presented. Finally, it is described how tool output was compared with measured data and numerical modelling for two estuaries filled with bars and three end-member systems (Figure 2), including the Western Scheldt and Columbia River. A general flowchart of the tool is given in Figure 3 and a list of symbols used is given in Appendix A Table A1. The full Python code of the tool and instructions on how to use it are available in the supplementary material and accessible on Github. The tool output comprises data exported in spreadsheet format and maps of bed elevation, inundation duration, peak flow velocity and salinity throughout the estuary.

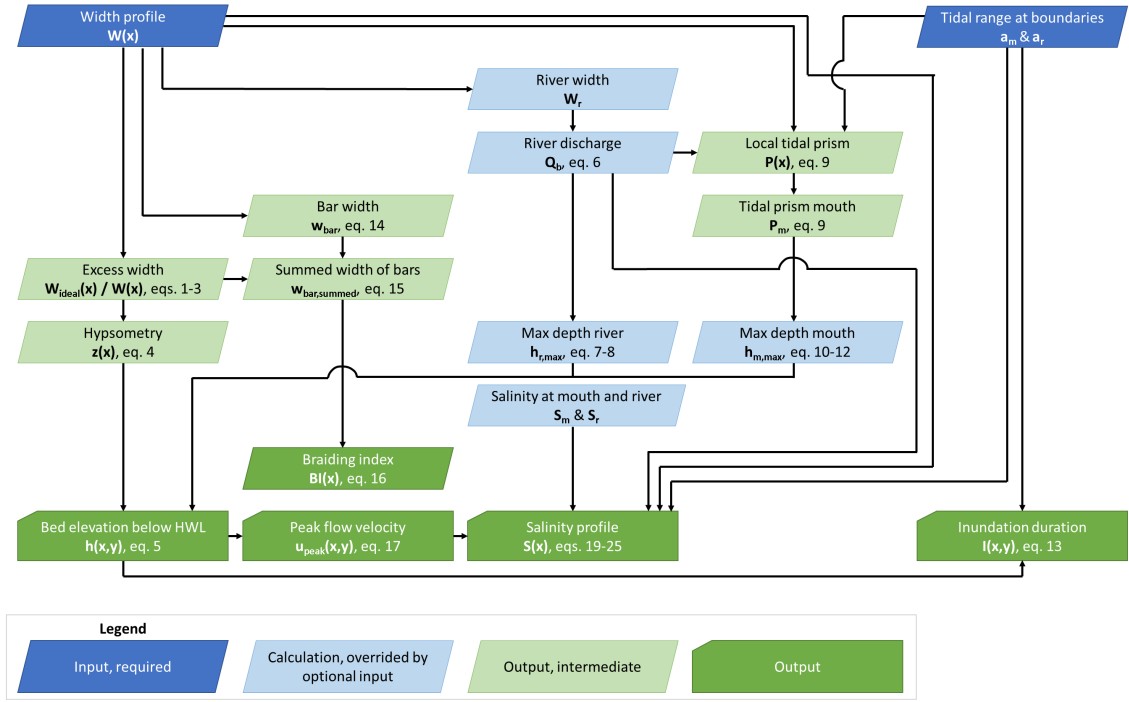

**Figure 3.** Flowchart of the calculation steps in the tool.

General assumptions underlying the entire method are as follows. There are no significant geological constraints on the depth along the estuary. At the landward and seaward boundary, we assume hydraulic geometry. Moreover, there is no significant effect of the offshore wave climate on the morphology of the mouth and bed elevations in the estuary. The along-channel tidal range is simplified as an along-channel profile that is constant or linearly increasing or decreasing. For calculations of inundation duration, a sinusoidal M2 tide is assumed.

### 2.1. Tool Approach

In general, the level of predictable detail depends on the available data. Data that are generally available for estuaries from all over the world are remotely-sensed imagery and a typical tidal range at the mouth, which means that our predictive relations should be based on and derivable from this information only. From remotely-sensed imagery, the along-channel width profile can be obtained (Section 2.2). It was previously found that the along-channel width profile correlates with the cross-sectional hypsometry [20]. The deviation from a perfectly converging planform translates in the occurrence of intertidal bars and therefore also in the shape of the hypsometric curve [33–35]. However, to scale dimensionless hypsometry to vertical bed elevation distributions, an along-channel depth profile is required. Estuary depth generally decreases linearly to near-linearly from the estuary mouth

to the upstream tidal limit citepSavenije2015, enabling channel-depth estimation at any location in the estuary when the depth at the mouth of the estuary and upstream river are known. For the landward boundary, hydraulic geometry relations for rivers predict channel depth from width or bankfull discharge [36,37], while the geometry of the estuary mouth is related to local tidal prism [13,15,17,38,39]. The typical tidal range at the mouth in combination with the along-channel width profile gives an estimate of the local tidal prism. This means that bed elevations and inundation duration are predictable from only the along-channel width profile and tidal amplitude profile, which is the minimum input data required to run the tool.

In order to be able to use additional data when available, we allow some variables to override estimations by the above relations (Figure 3). Specifically, measured maximum mouth depth and maximum depth at the upstream boundary make their estimation based on tidal prism and hydraulic geometry redundant. A specified depth and width at the upstream boundary implicitly impose a user defined hydraulic geometry. Similarly, measured fresh water discharge may override the estimation of discharge based on channel width, which replaces the high uncertainty of the empirical relation with the lower measurement uncertainty. Finally, salinity at the mouth and landward boundary may be specified instead of the default values of 35 ppt at the mouth and 0 ppt at the landward boundary.

## *2.2. Bed Elevation Prediction*

From remotely-sensed imagery, the first step is to obtain the along-channel width profile. Multiple options are available ranging from manually measuring the width at equally spaced transects to software tools that calculate river widths from remotely-sensed imagery (e.g., RivWidth [40,41]). Here, the following approach was used (following Leuven et al. [19]): first, we digitised the estuary planform in GIS software. Second, the polygon was imported in GIS software and the channel centreline was automatically extracted (e.g., Polygon To Centerline, ArcGIS function). Last, transects were drawn equally spaced on the centreline (e.g., Generate Transects Along Lines, ArcGIS function) and cropped with the planform polygon. The length of successive transects resulted in the along-channel width profile.

### 2.2.1. Estuary Shape Predicts Cross-Sectional Hypsometry

Estuaries typically narrow from the tidal inlet to the upstream river because the tidal wave generally dampens out in landward direction such that the combined discharge, or flow energy, of the river and the tides reduces. The estuary convergence is often characterised by a convergence length of width [18,42,43]. In ideal estuaries, the energy loss due to friction is balanced by channel convergence such that the tidal energy per unit of width remains constant up to the estuary head [17,44,45]. However, natural estuaries adapted in varying degrees to their initial and boundary conditions [9,27], resulting in planform shapes that deviate from the converging shape expected according to the theory for the ideal estuary state. The deviation in width from a converging shape was found to be a fairly accurate predictor of bar pattern and cross-sectional bed elevation distributions [19,20].

The convergence length ($L_W$) is the length over which the channel width reduces by a factor 2.72 (*e*) and is calculated as:

$$L_W = \frac{-s}{\ln\left(\frac{W_r}{W_m}\right)} \tag{1}$$

where $W_m$ is the width of the estuary mouth, $W_r$ is width of the river at the most landward boundary of the estuary and $s$ is the along-channel distance measured between the seaward and landward boundary. We followed the same guidelines on the selection of the landward and seaward boundary as in Leuven et al. [20], given on p. 766.

The ideal width ($W_{ideal}(x)$) and excess width ($W_{excess}(x)$) are assumed to be related to this convergence length and the local width ($W(x)$) of the estuary:

$$W_{ideal}(x) = W_m e^{\frac{-x}{L_W}} \tag{2}$$

$$W_{excess}(x) = W(x) - W_{ideal}(x) \tag{3}$$

Here, $x$ is the streamwise coordinate measured from the mouth along a centreline and $y$ is the coordinate perpendicular to the centreline.

Along-channel variations in excess width translate into bed elevation with cross-sectional hypsometric curves. Hypsometric curves describe bed elevations with a cumulative profile. Multiple empirical relations have been proposed for the hypsometric shape of (partially) submerged bodies [34,35,46,47] and terrestrial landscapes [33]. The original formulation of Strahler [33] (Equation (5)) appeared capable to describe hypsometries that occur in estuaries [20]. Two parameters in the Strahler [33] equation can be tuned to modify the hypsometric shape: $r$ sets the slope of curvature at the inflection point and $z$ sets the concavity of the function, with lower values representing a more convex cross-sectional bed profile and higher values representing a more concave profile.

Reasonably accurate predictions are obtained in estuaries with $r$ set to a constant value of 0.5, while the along-channel variation of best fitting $z$ strongly depends on the along-channel variation in width [20]. Moreover, at locations where the estuary is much wider than expected from an ideal shape, bars are typically more abundant, the hypsometry is more convex and $z$-values are higher. The along-channel variation in concavity ($z$) is given by:

$$z(x) = 1.4 \left( \frac{W_{ideal}(x)}{W(x)} \right)^{1.2} \tag{4}$$

The resulting predictions are accurate within a factor 2 of the measured value for intertidal and subtidal area in multiple estuaries of various size and character. This means that it is possible to predict the hypsometric shape per cross-section based on the estuary shape and, in this study, its predictive capacity for end-member cases is assessed as well.

The along-channel variation in concavity of the hypsometric shape ($z(x)$) can subsequently be used in the general hypsometric curve, formulated by Strahler [33] as:

$$h_z(x,y) = \left( \frac{r}{r-1} \right)^{z(x)} \left[ \frac{1}{(1-r)y+r} - 1 \right]^{z(x)} \tag{5}$$

in which $h_z(x,y)$ is the bed elevation above which proportion $y$ of the width profile occurs. Alternatively said, $h_z(x,y)$ is the proportion of total section height and $y$ the proportion of section width. $r$ is set to a constant value of 0.5 and $z(x)$ is given by Equation (4).

The bed elevation ($h_z(x,y)$) and the width fraction ($y$) in Equation (5) are dimensionless, where $h_z(x,y)$ is normalised between the local high water level (HWL) and the maximum estuary depth for that cross-section. Values for $y$ are dimensionalised with the local estuary width. Thus, to transform the predicted hypsometry to dimensional depth distributions, an along-channel profile of the maximum depth is required, which is estimated from the channel depth at the mouth and the depth at the landward boundary as explained in the next subsection.

2.2.2. Tidal Prism and River Discharge Predict Channel Geometry at the Mouth and Landward River

To calculate dimensional hypsometric profiles, an along-channel depth profile is required. Since the along-channel depth profile often shows a linear or almost linear profile [19,42], we fitted a linear along-channel profile on the maximum depth at the mouth of the estuary and the maximum depth at the landward river. When these values are known, the user can specify them, which overrides the

estimation. When unspecified, the maximum and average depth at the mouth and at the upstream river are estimated with hydraulic geometry, using respectively the tidal prism and river discharge.

Hydraulic geometry is an empirical construct and should be used with care. However, it is often the best estimate in the absence of observed data. The resulting predictions are dependent on the specific region or river type for which they were developed. Here, we implemented, and used, the equation Hey and Thorne [37] as an example. Nevertheless, users of the tool can specify a measured channel depth and river discharge as input when available because it overrides the calculation with Equations (6) and (7) and therefore implicitly impose a hydraulic geometry. Alternatively, the user can calculate these values with a hydraulic geometry equation suitable for the type of river considered or implement their own equation in the source code.

When unknown, river discharge is estimated from the channel width at the landward boundary with hydraulic geometry. Here, we used Hey and Thorne [37], rewritten as:

$$Q_b = \left( \frac{W_r}{3.67} \right)^{1/0.45} \tag{6}$$

in which $Q_b$ is the bankfull discharge and $W_r$ is the width at the upstream river, measured at a location with almost zero tidal influence.

Subsequently, the width-averaged depth at the landward boundary is estimated from bankfull discharge as:

$$\overline{h_r} = 0.33 Q_b^{0.35} \tag{7}$$

The maximum depth at the landward boundary is a function of the average depth and the geometric shape of the channel. Therefore, the following equation was adopted:

$$h_{max,r} = s_r \overline{h_r} \tag{8}$$

in which $s_r$ is the shape factor of the cross-section at the landward river, where 1 is a perfect rectangular and 2 a V-shaped cross-section. If unknown, a value of 1.85 is assumed for the landward boundary, based on geometry of five estuary mouths (Appendix A Figure A1).

The average depth at the mouth ($\overline{h_m}$) is calculated as the cross-sectional area at the mouth ($A_{csa,m}$, Equation (10)) divided by the local width ($W_m$), in which the cross-sectional area is estimated with the tidal prism. The local tidal prism is given as:

$$P(x) = \sum_{x}^{x+E} (2W(x)a(x)) + \frac{Q_b}{4} t \tag{9}$$

in which $P(x)$ is the local tidal prism resulting from the tidal amplitude ($a(x)$) and surface area upstream of location $x$. The contribution of river discharge ($Q_r$) to the local tidal prism is approximated as the bankfull discharge ($Q_b$) divided by 4. $E$ is the tidal excursion length, which is the distance a water particle travels over half a tidal cycle. Since $E$ is dependent on flow velocities, which are unknown at this point in the assessment tool, $E$ has to be estimated at this point. The tidal excursion length is estimated by multiplying a typical tidal flow velocity with $t$, which is the duration of half a tidal period. According to Savenije [18], the order of magnitude of the tidal excursion length can be estimated even in ungauged estuaries because the peak velocity amplitude in alluvial estuaries from all over the world is remarkably similar, being approximately $1 \text{ ms}^{-1}$. Here, we use this typical flow velocity to obtain an estimate of the tidal excursion length, which will be evaluated in the results. In Leuven et al. [19], the upper limit for summation of the tidal prism was the upstream estuary boundary, but given that the input of the tool may comprise estuaries much longer than the tidal excursion length, the summation is here limited to the tidal excursion length.

Subsequently, the cross-sectional area at the mouth is related to the tidal prism. Earlier approaches derived these relations for tidal inlets or specific tidal systems [13,15,38,39,48–50]. Recently, we derived

an empirical relation for multiple cross-sections within 35 estuaries (supplementary material of [19]). This equation is used to estimate the cross-sectional area at the mouth:

$$A_{csa}(x) = 0.13 \times 10^{-3} P(x) \tag{10}$$

in which $A_{csa}(x)$ is the along-channel cross-sectional area.

Now, the average depth at the mouth is calculated as:

$$\overline{h_m} = \frac{A_{csa,m}}{W_m} \tag{11}$$

and converted to a maximum depth at the mouth with

$$h_{max,m} = s_m \overline{h_m} \tag{12}$$

in which $s_m$ is the shape factor of the cross-section at mouth. If unknown, a value of 1.65 is suggested (Appendix A Figure A1).

### 2.3. Inundation Duration Prediction

Dimensional depth distributions directly translate in inundation durations assuming a simple harmonic tide. Dimensional depth profiles are obtained from scaling the cross-sectional hypsometric profiles with the along-channel profile of maximum depth. Bed elevations below the low water level (below $-1$ times the amplitude) are always submerged and are therefore assigned a relative inundation duration of 1. For the intertidal zone, the inundation duration was calculated as:

$$I(x,y) = 0.5 - 0.5 \sin\left(\frac{0.5\pi}{a(x)} h_z(x,y)\right) \tag{13}$$

### 2.4. Bar Pattern Prediction: Bar Width and Braiding Index

The along-channel variation in bar pattern correlates with along-channel variations in width. Moreover, the summed width of bars is equal to the excess width and thus Braiding Index ($BI(x)$) can subsequently be calculated by dividing the excess width ($W_{excess}(x)$) by the predicted bar width ($w_{bar}(x)$) [19]:

$$w_{bar}(x) = 0.39 W(x)^{0.92} \tag{14}$$

$$w_{barssummed}(x) = W_{excess}(x) \tag{15}$$

$$BI(x) = \frac{w_{barssummed}(x)}{w_{bar}(x)} \tag{16}$$

### 2.5. Flow Velocity Prediction

A relation for typical flow conditions as a function of estuarine geometry was still lacking. Therefore, we explored this relation in a calibrated hydrodynamic model of Rijkswaterstaat (SCALWEST, [32]) for the bathymetry of the Western Scheldt in 2009. We hypothesised that a relation would exist between bed elevation and typical flow velocity (Figure 1b). Peak tidal flow velocity ($u_{peak}$) indeed showed a good correlation ($R^2 = 0.73$) with depth below high water level (Figure 4):

$$u_{peak}(x,y) = 0.10 + 0.91_{10}\log(h(x,y)) \tag{17}$$

Typical average flow velocity over half a tidal cycle was estimated depending on the deviation from the maximum along-channel tidal amplitude as follows (Appendix A Figure A5):

$$u_{avg}(x,y) = u_{peak}(x,y)\left[1 - \frac{a(x)}{\max(a(x))}\frac{2}{\pi}\right] \tag{18}$$

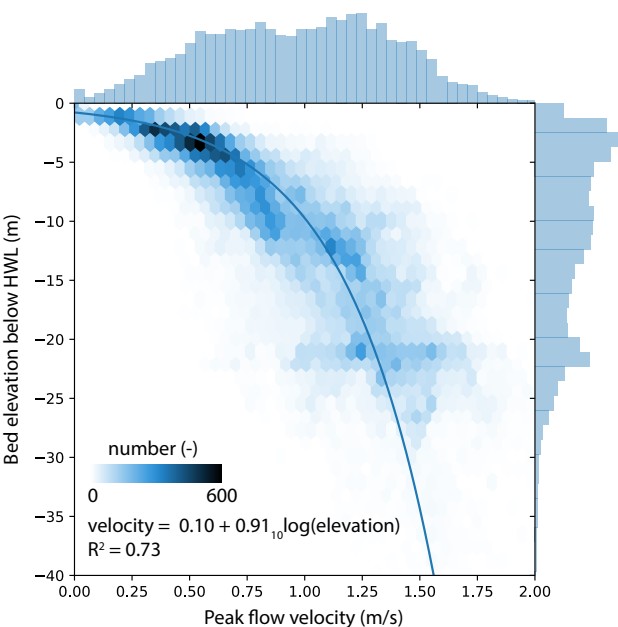

**Figure 4.** Relation between peak tidal flow velocity and depth below high water level from the Western Scheldt model (Figure 1a,b). Histograms indicate occurrence within bins and colours in the plot become more saturated with increasing point density.

### 2.6. Salinity Prediction

Salinity along the estuary depends on the freshwater discharge at the upstream boundary and the degree of mixing by the tides. Three along-channel salinity predictors were implemented in the assessment tool [21–23] and the quality of these predictors for the cases studied here will be described in the results.

Savenije [21] proposed a one-dimensional model that uses predictive equations for mixing and dispersion, such that it can be applied to systems for which limited information is available. The resulting model has been tested on a set of 15 estuaries. Gisen et al. [23] revised the predictive equation of Savenije [21], calibrated the model on 20 estuaries and validated it with another 10 estuaries. Brockway et al. [22] proposes a solution for advection-diffusion equations in converging estuaries and implements an empirical relation for longitudinal mixing based on data from the Incomati Estuary (Mozambique).

The output maps in the tool are based on the averaged prediction of Savenije [21] and Gisen et al. [23]. Nevertheless, the output for Brockway et al. [22] is also provided. The along-channel salinity profile of Brockway et al. [22] is typically exponentially decreasing to a salinity of 0, while the other two are generally more s-shaped with a strong gradient in the middle of the curve. Taking the average of all three predictors would thus not make sense because their characteristic shape would smoothen out. First, the latter is described because it has a simpler solution and less constitutive variables.

The salinity profile of Brockway et al. [22] ($S_b(x)$) is given as:

$$S_b(x) = S_m \exp \left\{ \frac{-Q_r}{L_{A_{csa}}^{-1} K A_{csa,m}} \left[ e^{(L_{A_{csa}}^{-1}x)} - 1 \right] \right\} \tag{19}$$

in which $S_m$ is the salinity at the mouth of the estuary (35 ppt), $A_{csa,m}$ is the cross-sectional area at the mouth, $L_{A_{csa}}$ is the convergence length of the cross-sectional area and $K$ is an empirical equation for the longitudinal mixing coefficient based on average river discharge ($Q_r$) and tidal range ($a(x)$), being $K = 0.28Q_r + 13\overline{a(x)}$.

The salinity profile of Savenije [21] ($S_s(x)$) is given as:

$$S_s(x) = (S_m - S_r) \left( \frac{D(x)}{D_m} \right)^{1/K} + S_r \tag{20}$$

in which $S_r$ is the salinity at the fresh water river (0 ppt), $K$ is the longitudinal mixing (Van der Burgh) coefficient (typically 0.07), $D(x)$ is the dispersion coefficient and $D_m$ is the dispersion at the mouth of the estuary (typically 7000 m²s⁻¹). The ratio between the along-channel dispersion and dispersion at the mouth is given in Savenije [21] as:

$$\frac{D(x)}{D_m} = 1 - \left( \frac{K L_{A_{csa}} Q_r}{D_m A_{csa,m}} \right) \left( e^{\frac{x}{L_{A_{csa}}}} - 1 \right) \tag{21}$$

with

$$D_m = 220 \sqrt{40} \frac{\overline{h_m}}{L_{A_{csa}}} \sqrt{N_r} \max(u_{peak}(mouth, y)) E \tag{22}$$

in which

$$N_r = \frac{Q_r t / P(m)}{\rho_w \max(u_{peak}(mouth, y))^2 / \Delta \rho g \overline{h_m}} \tag{23}$$

where $\rho_w$ is the density of water, $\Delta \rho$ is the density difference between salt and fresh water and $g$ is the gravitational acceleration (9.81 ms⁻²).

The salinity profile of Gisen et al. [23] ($S_g(x)$) is equal to the profile of Savenije [21] for the part landward of the inflection point, which is the part used in this study, except for the equation for $K$. The longitudinal mixing coefficient ($K$) in Savenije [21] ($K_s$) is given as:

$$K_s = 0.16 \times 10^{-6} \frac{\overline{h_m}^{0.69} g^{1.12} t^{2.24}}{(2a(m))^{0.59} L_W^{1.1} W_m^{0.13}} \tag{24}$$

while, in Gisen et al. [23], it ($K_g$) is given as:

$$K_g = 151.35 \times 10^{-6} \frac{W_r^{0.3} (2a(m))^{0.13} t^{0.97}}{W_m^{0.30} C_m^{0.18} [\max(u_{peak}(mouth, y))]^{0.71} L_W^{0.11} \overline{h_m}^{-0.15} r_s^{0.84}} \tag{25}$$

in which $a(m)$ is the tidal amplitude at the mouth, $C_m$ is the Chezy roughness, estimated in the assessment tool as 42 m⁰·⁵s⁻¹, which is adjustable in the tool code, $\max(u_{peak}(mouth, y))$ is the peak tidal flow velocity in the cross-section at the mouth, and $r_s$ is the storage ratio, calculated as the width of intertidal area at the mouth divided by the width of the subtidal area at the mouth. The minimum value of $K$ is 0 and the maximum value is 1.

## 2.7. Tool Validation

For purposes of illustration and validation, the tool was applied to the Columbia River estuary (USA), which is an example of a relatively wide valley that is partially filled with bars [26], and the Western Scheldt (NL), which is an alluvial estuary with large deviations from a converging along-channel width profile and filled with tidal bars [19,26] (Figure 2c,d). For validation, channel depth at the landward boundary was calculated in the tool with measured river discharge. Predicted velocities are validated on the Columbia River only because the regression was based on data for the Western Scheldt. Bed elevation and salinity predictions are validated on both systems. The predicted flow velocities from the tool were compared with results from numerical modelling for the Columbia River estuary [7,28] and from a calibrated hydrodynamic model of Rijkswaterstaat for the Western Scheldt (SCALWEST, [32]). Results from numerical modelling were used because field measurements of flow velocities do not provide full coverage in time and space. Salinity data were obtained from

Jay and Smith [51] for the Columbia River and from de Brauwere et al. [52] and Vroom et al. [53] for the Western Scheldt. Habitat preferences for potential habitat maps were obtained from the Marine Life Information Network (MarLIN), which is a database on the biology of species and the ecology of habitats.

### 2.8. Tool Application to End-Member Cases

We tested the tool's performance against end-member systems. The dominant factors controlling the tool output are (i) the along-channel width profile, (ii) river discharge at the landward boundary and (iii) tidal range at the seaward boundary. Given these three factors, we designed the three most extreme end-member cases (Figure 2a,b,e), which were tested in addition to two estuarine cases with variable width profile (the Western Scheldt and Columbia River) (Figure 2c,d). The first end-member is an estuary with a perfectly converging shape from the mouth to the upstream river, i.e., negligible width variation. This case was based on the dimensions of the Tran De branch of the Mekong Delta (Vietnam). The second end-member is a river with a constant channel width, i.e., no tidal range at the seaward boundary and negligible width variation. This case was based on the dimensions of a distributary of the Saskatchewan River, near Cumberland Lake (Saskatchewan, Canada). The third end-member is a tidal basin, i.e., no river discharge and landward increasing width instead of decreasing. This case was based on dimensions of the Ameland inlet and represented by a narrow mouth, an increasing channel width in landward direction and an infinitely small river at the landward boundary. It should be noted that, in the last case, along-channel width is measured perpendicular to the tidal channels, which means that the transects for an idealised tidal basin are semicircles. The predictions for the Tran De branch were compared with measured data [54–56] and the predictions for the tidal basin were compared with numerical model results [31,57].

## 3. Results

### 3.1. Tool Output

The tool output comprises maps of predicted bed elevation with respect to the high water level (HWL), relative inundation duration, flow velocity and salinity (Figures 5 and 6). In general, the bed deepens in seaward direction. At locations where the estuary is relatively wide, average bed elevations are higher and at locations where the estuary width is close to the ideal width, bed elevations are lower (Equations (4) and (5)). Zones where the excess width is large are the locations where stretches of intertidal area are wider and where inundation durations are less than 100%. Peak flow velocities increase in seaward direction with maxima around 1.8 $\mathrm{ms}^{-1}$ at the mouth. Peak flow velocities in the intertidal area are generally below 0.8 $\mathrm{ms}^{-1}$. The salinity gradually decreases in landward directions and approaches river salinity at the landward boundary.

Before, we had to estimate a typical peak flow velocity because it was required for the estimation of the tidal excursion length (Equation (9)). At that point, a value of 1 $\mathrm{ms}^{-1}$ was assumed [18]. This assumption only partly influences the output flow velocity, which depends on bed elevation, which in turn is controlled by the estuary planform shape, the tidal range and the tidal excursion length. Nevertheless, output maps justify the assumption of 1 $\mathrm{ms}^{-1}$ for the purpose of calculating the tidal excursion length. The average peak tidal flow velocity is 1 $\mathrm{ms}^{-1}$ for the Western Scheldt and 1.2 $\mathrm{ms}^{-1}$ for the Columbia River when averaged over the surface area of a typical tidal excursion length (Figures 5 and 6). Median values of peak tidal flow velocity per transect are also remarkably close to the typical value proposed by Savenije [18] along the entire estuary (Figure 7a,c).

Additional output (Appendix A Figures A3 and A4) consists of predictions of bar width and braiding index per cross-section, maximum peak flow velocity and maximum average flow velocity per transect. Furthermore, four zones are calculated based on elevation with respect to the tidal range: subtidal deep ($<-2a$), subtidal low, intertidal low (more than 50% of time submerged) and intertidal

high (less than 50% of time submerged). For each of these zones, the width of the zone, average depth and average and peak tidal flow velocities are calculated as output.

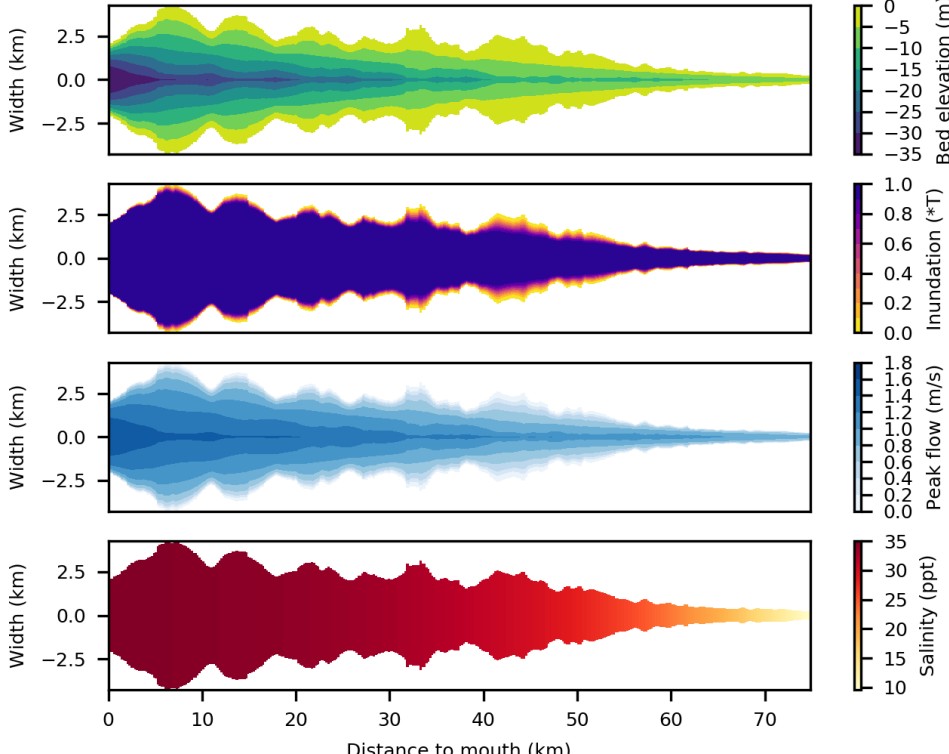

**Figure 5.** Resulting predictions of bed elevation with respect to the high water level, inundation duration, peak flow velocity and salinity for the Western Scheldt.

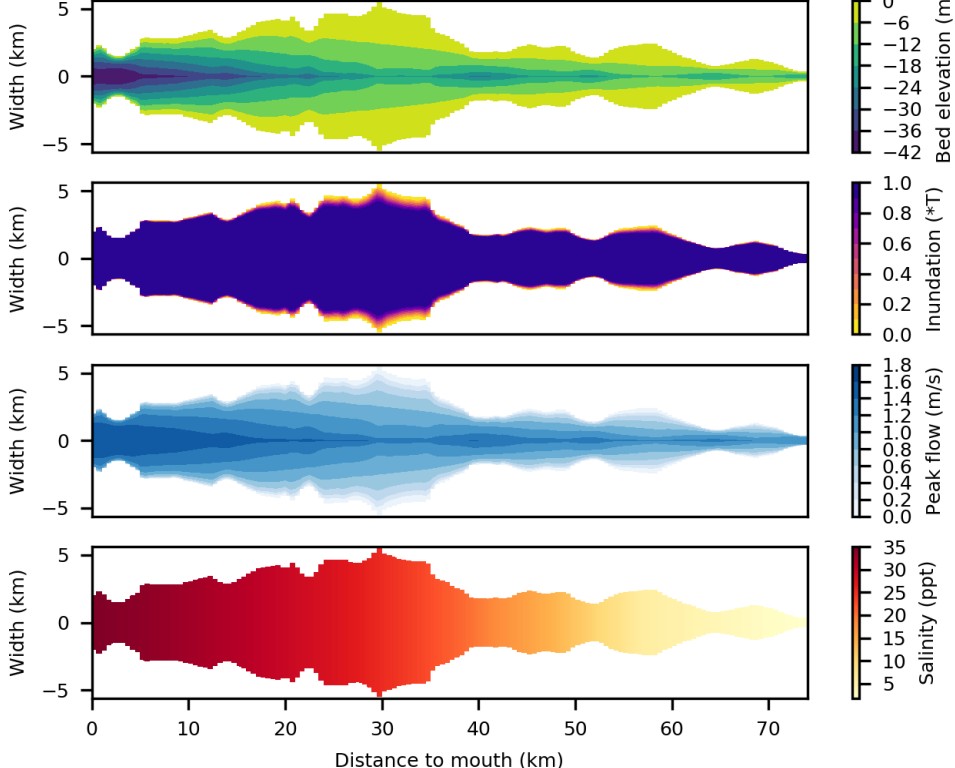

**Figure 6.** Resulting predictions of bed elevation with respect to the high water level, inundation duration, peak flow velocity and salinity for the Columbia River.

*3.2. Tool Validation*

3.2.1. Bed Elevations

To validate the predicted bed elevations, median and minimum values per transect were compared with measurements from bathymetry. Median and minimum bed elevations are generally within a factor 2 of the measured values from bathymetry (Figure 7a,b). Median bed elevations deviate the most in the landward part for the Western Scheldt (Figure 7a). In this section, a larger portion of the width is dredged, which causes the median depth to fall within the deeper dredged channel. The deviation between predicted and measured values lacks a trend with a position along the estuary (as indicated by the colours in Figure 7a,b). The largest outliers for the Columbia River are found at the mouth where the predicted median bed elevation is much lower than measured from bathymetry (Figure 7b). This is caused by the small embayments enclosed by groynes, located on the sides of the mouth area. In this zone, a larger proportion of high bed elevations causes a higher median bed elevation.

3.2.2. Flow Velocity

The range of predicted flow velocities is generally in the same range as the modelled values for both the peak and average flow velocities (Figure 7c,d). However, the along-channel variation in minimum and maximum flow velocities is larger for the models than for the tool output (Figure 8). We explain this by a simplification of some morphological elements in the tool. For example, the presence and location of scours and sills, which cause variation in the along-channel maximum depth, is not predicted by the tool. However, they do occur in natural systems and therefore also cause variations in along-channel flow velocity pattern. In the tool, the profile of maximum depth is linear along-channel, which results in lower variations in the maximum flow velocity. Nevertheless, the velocity predictions do capture the along-channel variability caused by the large-scale morphological variation in bed elevation, as illustrated with the median values per transect (Figure 8).

The tool does not reproduce the highest observed flow velocities in the Columbia River, which are well above 2 ms$^{-1}$ in the first 15 km from the mouth. To obtain velocities above 2 ms$^{-1}$ in the tool, bed elevation of at least 120 m below high water level are required (Equation (17)), which generally do not occur in natural systems. This means that the tool underpredicts flow velocities in cases where the cross-sectional area is relatively small compared to the local tidal prism, which might also suggest disequilibrium. Moreover, the regression between flow velocity and depth, simplifies the increasing scatter for larger depths and flow velocities, which is observed for the Western Scheldt (Figure 4).

Predicted velocity against modelled velocity shows a trend for the Columbia River with modelled velocities below 1 ms$^{-1}$ being overpredicted by the tool and velocities above 1 ms$^{-1}$ being underpredicted (Figure 7d). The degree of misprediction does not correlate with position along the estuary. The only explanation for the deviation is a different depth dependency of peak flow velocity caused by a different ratio between tidal prism and cross-sectional area.

Therefore, an alternative approach for calculating flow velocities was based on the local tidal prism and the cross-sectional channel geometry, i.e., continuity. For each cross-section, a typical velocity is calculated at the average depth of the cross-section (i.e., $P(x)/(\overline{Th(x)})$). Nevertheless, the quality of these predictions appeared to be less (Figure 9) for two reasons: (1) at the upstream boundary, velocities are mainly sensitive to river discharge. Here, local tidal prism approaches river discharge because it is obtained from integrating over the distance of a tidal excursion length upstream (Equation (9)) plus river discharge. In the case of the Columbia River, this leads to overprediction of velocities because river discharge is large, while, for the Western Scheldt, it would lead to underpredictions. (2) Flow velocities at the intertidal area and deepest parts still have to be extrapolated from the average prediction creating uncertainty in the most important parts for respectively ecology and shipping.

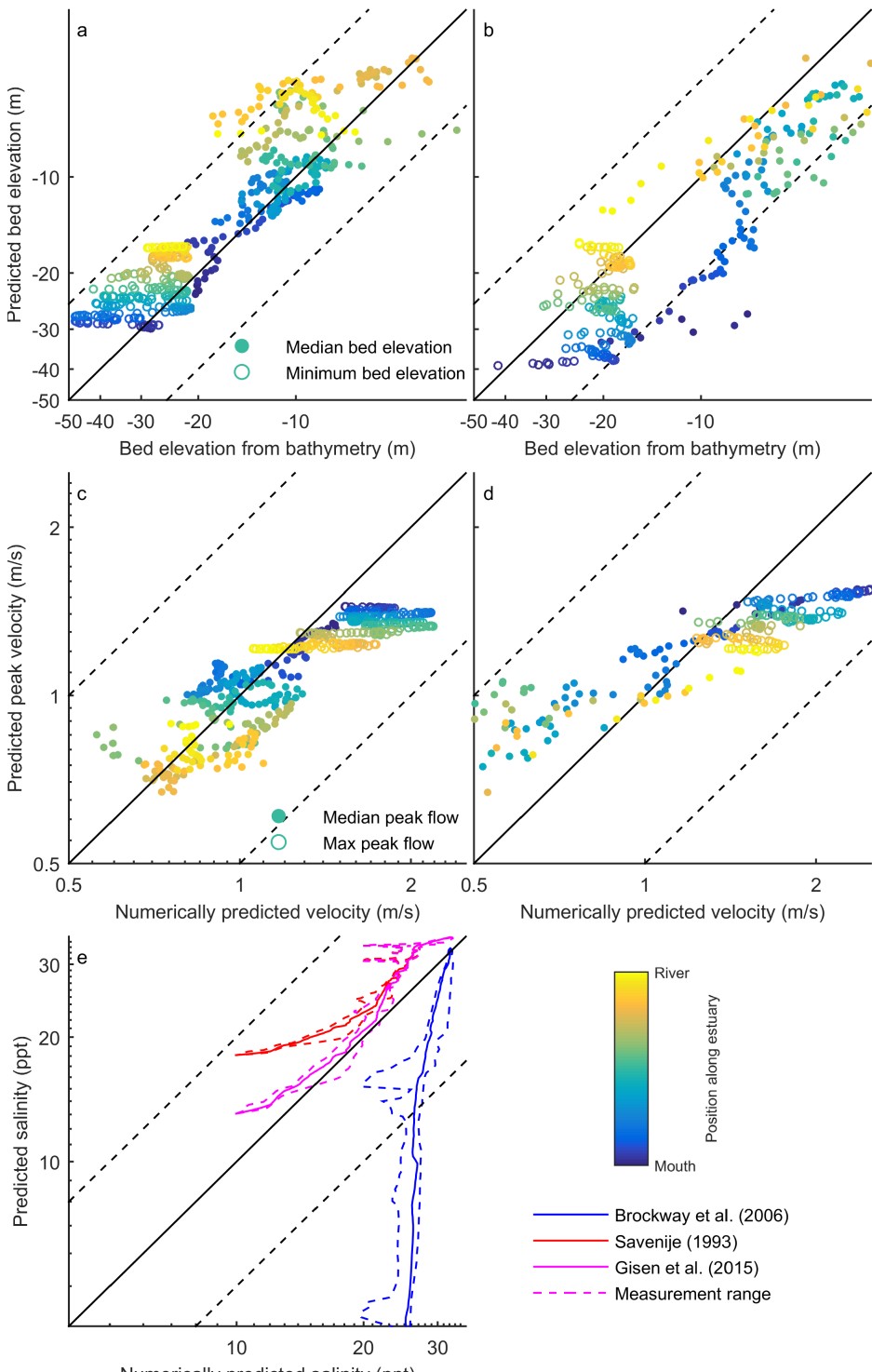

**Figure 7.** (**a**,**b**) predicted bed elevations against measured bed elevations taken from bathymetry for the Western Scheldt (**a**) and Columbia River (**b**); (**c**,**d**) predicted flow velocity from the tool against modelled flow velocity for numerical models of the Western Scheldt (**c**) and Columbia River (**d**). Filled circles indicate median values per cross-section, open circles indicate maximum values per cross-section. Dashed lines indicate a spread of a factor 2 around the $x = y$ line. Colours indicate the position along the estuary, with blue being the mouth and yellow being the most landward location; (**e**) predicted salinity for the three predictors against measured salinity in the Western Scheldt. Solid lines indicate mean measured salinity per transect against predicted salinity and dashed lines indicate maximum and minimum measured values.

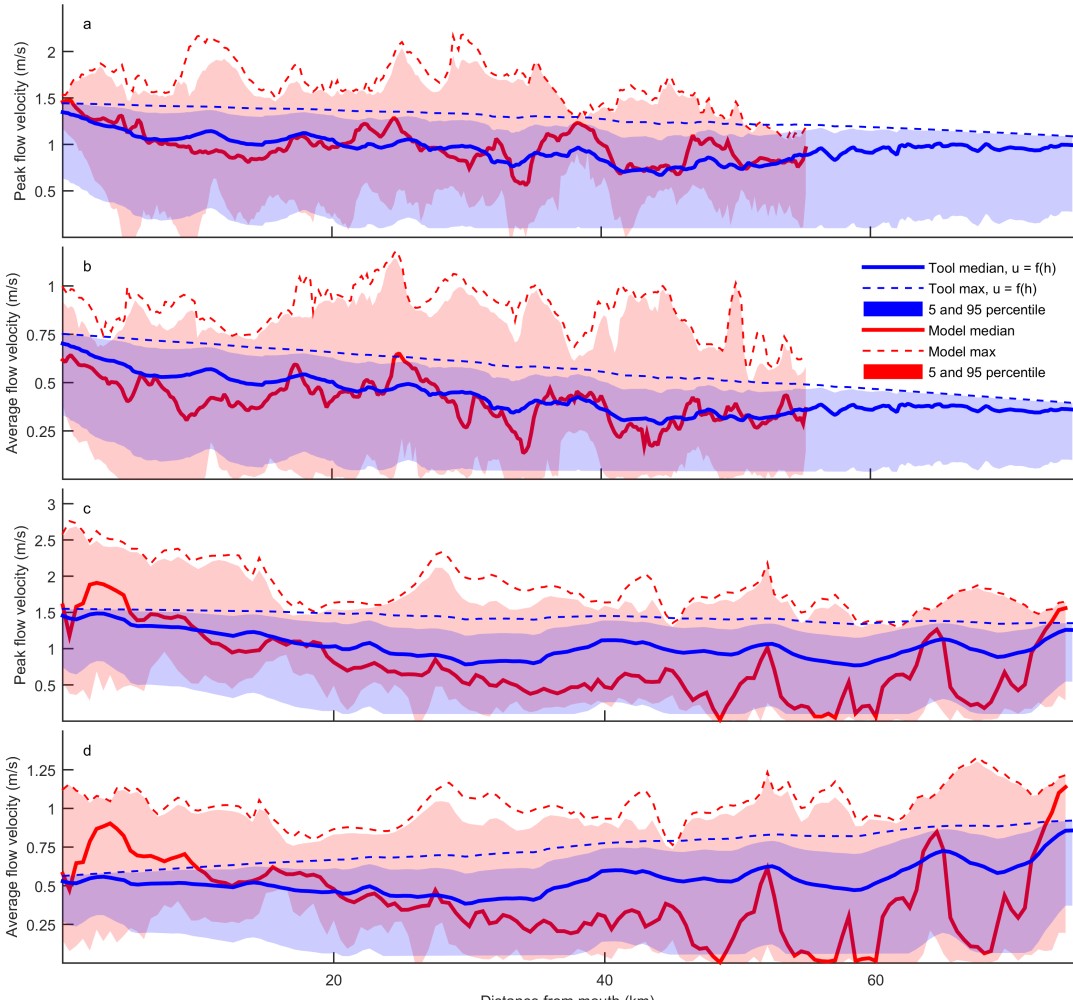

**Figure 8.** Comparison between modelled [red] and predicted [blue] flow velocities along the Western Scheldt (**a**,**b**) and the Columbia River (**c**,**d**). Shaded area shows the range of peak velocities (**a**,**c**) and range of average velocities over half a tidal cycle (**b**,**d**) bounded by the 5 and 95 percentile values. The solid line indicates the median and dashed line maximum value per transect.

Following this alternative approach, we anticipated that there would be a relation between the width variation and the misprediction based on depth only (Equation (17)). It was expected that a local increase in width would lead to flow expansion and therefore typically lower flow velocities than for a cross-section with similar tidal prism but lower width. However, the lack of trend between velocity deviation and width measures (Appendix A Figure A2) indicates that velocity predictions will not improve from including width in the regression.

The alternative approach shows that a prediction of flow velocities depending on tidal prism and depth does not reproduce modelled flow velocities better (Figure 9a,b,e,f). A predictive relation between flow velocity and depth is thus of value. This urges the need to compile a dataset with flow velocities and water depths of multiple systems. From that data, a relation that is better applicable to more systems could be derived, but this data is not yet available. Nevertheless, the present relation predicted flow velocity within a factor 2 of the modelled values even in the most extreme cases, which means it is of value for an estimate when limited data is available.

Predicted flow velocity and local depth result in a local tidal prism that is in the same range as the modelled values (Figure 10). Local tidal prism obtained from integrating a tidal range over the full length of the estuary is typically 1.5–2 times larger at the mouth than the tidal prism obtained from integrating over a typical tidal excursion length. This explains why using the latter is required to

obtain accurate predictions of the depth at the estuary mouth and additionally is less sensitive to the length of the estuary.

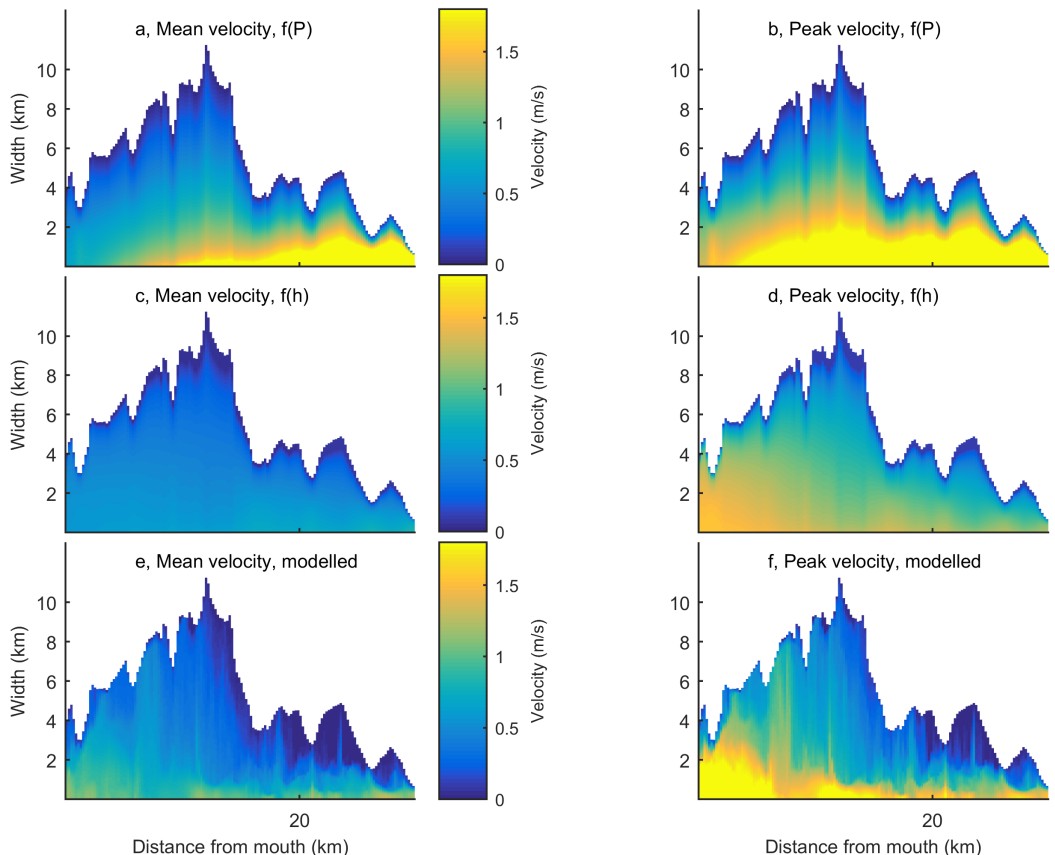

**Figure 9.** Mean velocity over a tidal cycle (left) and peak tidal flow velocity (right) as a function of local tidal prism (**a,b**), bed elevation (**c,d**) and numerically modelled values (**e,f**) for the Columbia River.

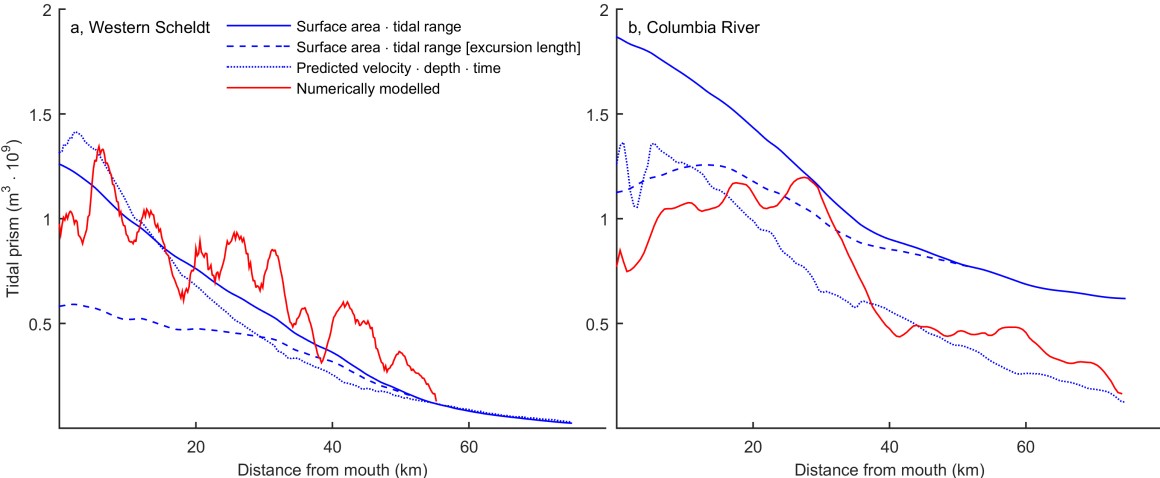

**Figure 10.** Resulting predictions of local tidal prism for the Western Scheldt (**a**) and Columbia River (**b**). Red lines are model results, blue lines are tool output.

### 3.2.3. Salinity

Predicted salinities are within the range of measured values [51] and values obtained with a numerical model with shallow-water and tracer-transport equations [52]. For the Western Scheldt, local salinity can vary between 4–8 ppt over a tidal cycle, with averages of 28 ppt at 10 km, 22 ppt

at 30 km and 16 ppt at 45 km from the mouth [52]. In the Columbia River estuary, temporal salinity variation is even larger [51], with local variations of 20–25 ppt in the most seaward 15 km. Predicted salinities fall within the measured range, but salinities are overpredicted in the most landward reach, where measured values approach 0 ppt from approximately 45 km from the mouth [51].

The salinity predictor of Gisen and Savenije [50] is most accurate for the salinities observed in the Western Scheldt (Figure 7e), which was a system used for validation in Gisen and Savenije [50] and not for calibration. This is the only salinity predictor for which predicted salinity is within a factor 1.5 of the measured value for the entire along-channel section. Predictors of Savenije [21] and Brockway et al. [22] show large deviations at the landward side, where the predicted salinity gradient is respectively too weak and too strong compared to measured values (Figure 7e).

### 3.3. Applicability to End-Member Systems: A River, an Ideal Estuary, and a Tidal Basin

In the case of a river and ideal estuary, the local channel width will always be equal to the ideal channel width as obtained from a maximum fitting converging shape. This means that the shape of the cross-sectional hypsometry is concave and constant along-channel (Equations (4) and (5), Figure 11). The concave hypsometry results in a relatively constant along-channel bed elevation profile (Figure 11). Only the along-channel variation in maximum depth and increase in channel width in seaward direction cause a slight along-channel variation. The implication of an along-channel constant concave hypsometry is that intertidal area is very small and inundation duration is 100% for the largest part of the estuary. Typical flow velocities are also along-channel constant at approximately 1–1.2 ms$^{-1}$. A strong salinity gradient is predicted at the mouth of the estuary, with salinities dropping to ≈5 ppt at 30 km from the mouth.

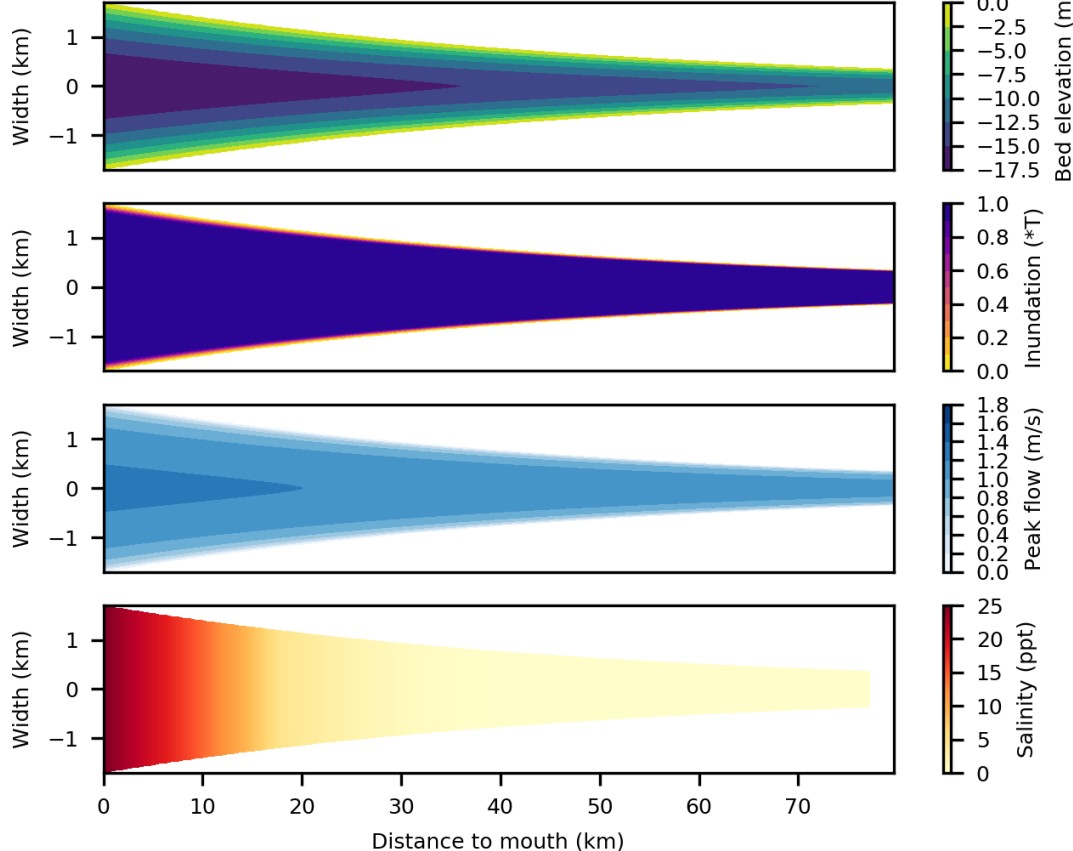

**Figure 11.** Resulting predictions of bed elevation with respect to the high water level, inundation duration, peak flow velocity and salinity for an ideal estuary based on dimensions of the Tran De branch of the Mekong river.

Ideal estuaries are said to be characterised by an along-channel constant velocity amplitude and depth [18,54] and these features are reproduced by the tool predictions. Moreover, the predictions correspond to observations in the Tran De branch of the Mekong: observed peak flow velocity is approximately 1 ms$^{-1}$ along the entire branch [55,56] and salinity intrusion reaches up to 45–55 km from the mouth [54]. In addition, the along-channel predicted salinities fall within the range of observed salinities [54]. The only feature not completely reproduced is the along-channel constant depth. When the maximum depth at the mouth is calculated from tidal prism, predicted depth at the mouth is a factor 1.5 larger, 5 m deeper, than observed.

The prediction for depth at the mouth leads to a misprediction at the most downstream boundary for the river (Appendix A Figure A6). In that case, maximum depth at the landward boundary is controlled by river discharge (to the power of $\approx$0.35), the width of the river without tidal influence (to the power of $\approx$0.77) or if both of these are unspecified, the channel width at the landward boundary. Depth at the mouth is calculated with the tidal prism, which in the case of a river is equal to the river discharge times the duration of half a tidal cycle. Therefore, depth at the mouth is also a function of river discharge or landward river width, but to the power of $\approx$1. This means that the predicted channel depth at the mouth will always be deeper than at the upstream boundary, especially in the end-member case of a straight river.

An ideal tidal basin is characterised by a strong landward increase in width, while the ideal width approaches 0 at the landward end. This means that the ratio of $W_{ideal}(x)$ to $W(x)$ strongly decreases in landward direction, resulting in convex hypsometry (Equation (4)) and therefore large amounts of intertidal area within the basin (Figure A7). In this end-member case, the depth at the mouth is set by the surface area of the tidal basin multiplied with the tidal range. In the case of a tidal basin based on the Ameland inlet, the tool predicts a maximum depth at the mouth of 20 m below the high water level. This seems appropriate when compared with modelled [31,57] and measured values [29] (Figure A8). Flow velocities at the mouth are in the order of 1 ms$^{-1}$, which is comparable with measured and modelled values for tidal basins in equilibrium [57,58].

## 4. Discussion

### 4.1. Sensitivity of Tool Output to Tidal Range, River Discharge, and along-Channel Width Profile

The sensitivity of the input tidal range, river discharge, and along-channel width profile was tested on the following parameters: (1) the ratio between intertidal and subtidal area; (2) the ratio between low and high dynamic environment; and (3) peak flow velocity at the estuary mouth (Figure 12). Low dynamic environment, which forms potential habitat, is defined as intertidal area where flow velocities are below 0.8 ms$^{-1}$. All other areas are classified as a high dynamic area.

While most sensitive dependent variables are increasing or decreasing monotonously with the independent variable, the ratio of low and high dynamic intertidal area as a function of tidal amplitude shows an optimum. An increase in tidal amplitude increases the ratio between intertidal and subtidal area (Figure 12a) because the increased tidal range increases intertidal area. However, increases in tidal amplitude also generate a proportional increase in tidal prism. A larger tidal prism at the mouth will increase the predicted cross-sectional area and because the width is fixed; this will increase predicted depth at the mouth. Because maximum depth along the estuary is dependent on the maximum depth at the mouth and the maximum depth at the landward river, this increases the maximum depth along the entire estuary. The increased depths subsequently increase the predicted flow velocities (Figure 12g). This means that two opposing mechanisms affect the ratio between low dynamic and high dynamic environment: increased tidal range increases intertidal area which forms a more low-dynamic environment. In contrast, the deepening by increase in tidal prism increases flow velocities and therefore reduces the low dynamic area. Therefore, the combined effect results in an optimum curve for the ratio between low dynamic (velocity < 0.8 ms$^{-1}$) intertidal area and high dynamic area as a function of tidal range (Figure 12d).

An increase in the river discharge results in less intertidal area, less low dynamic area and larger peak flow velocities (Figure 12b,e,h). After all, increased river discharge increases the tidal prism, which increases channel depths.

The ratio between ideal width and excess width affects the distribution of bed elevations per cross-section and the tidal prism. In estuaries that adapt to the ideal shape, the excess width becomes lower. This results in more concave cross-sectional hypsometry and therefore narrower stretches of intertidal flats (Figure 12c). The tidal prism decreases as a result of reduced estuary area, resulting in lower maximum depths and therefore also lower flow peak velocities (Figure 12i).

A comparison between predicted and measured or modelled values showed that bed elevations, flow velocity and salinity can be predicted within a factor two of the measured or modelled values (Figure 7). The largest deviations between predicted and modelled flow velocities occur when they peak above 2 ms$^{-1}$ in natural systems. Largest mispredictions for salinity occur at the transition from brackish to fresh water, where salinity is also the most dynamic over the seasons. Bed elevation predictions are most uncertain when the tidal range is low for the depth at the mouth, which is also sensitive to offshore wave climate that we have not considered here. This implies that the tool is capable to provide a first-order estimation for estuaries with limited data, but that it cannot replace measurements or numerical modelling.

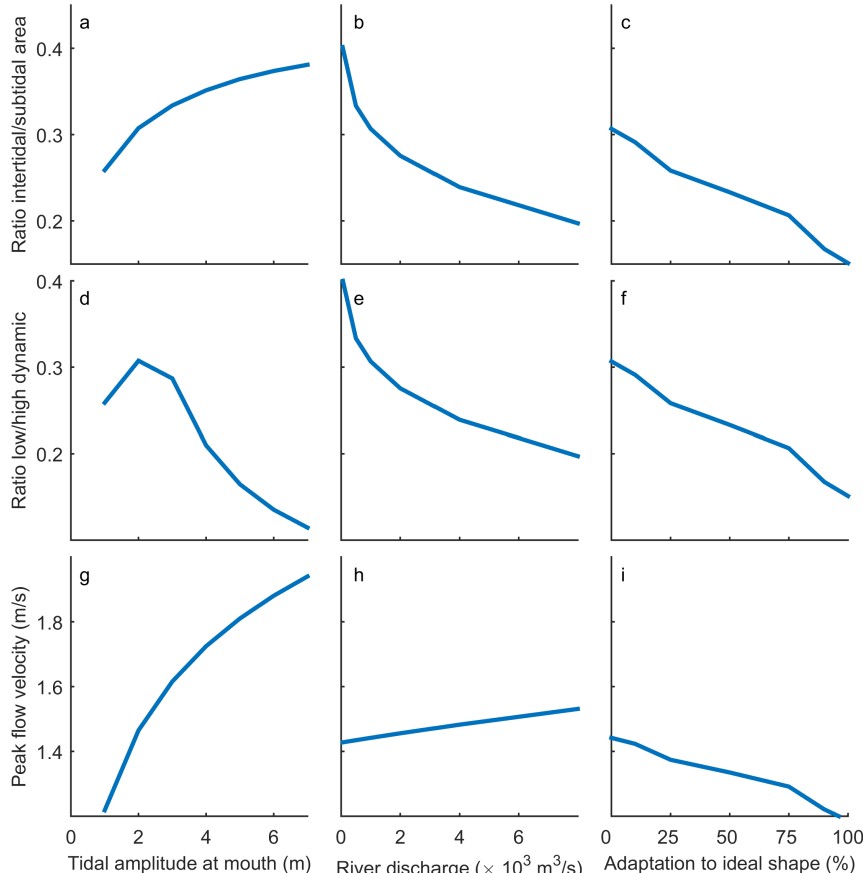

**Figure 12.** Sensitivity of tool output for the Western Scheldt to input tidal amplitude, river discharge and width profile. The ratio between intertidal and subtidal area increases with tidal amplitude (**a**) and decreases with river discharge and adaptation to ideal shape (**b**,**c**). The ratio between low dynamic (velocity $< 0.8$ ms$^{-1}$) intertidal area and high dynamic area has an optimum curve for tidal amplitude (**d**) and decreases with river discharge and adaptation to ideal shape (**e**,**f**). Peak flow velocity at the mouth of the estuary increases with tidal amplitude and river discharge (**g**,**h**) and decreases with adaptation to ideal shape (**i**).

*4.2. Applications*

4.2.1. Illustration of Application for Ecological Assessment

The major abiotic factors that determine habitat area of living organisms in estuaries are depth, flow velocity, salinity and inundation time [2–4,59]. Habitat suitability models (e.g., HABITAT, [24]) can translate these abiotic factors into a characteristic habitat suitability index, which for example has been applied to study the effect of climate change and dam construction on potential habitat in rivers [25]. Multiple case studies of individual systems or individual species are also available for estuaries [60–64], but application to more systems worldwide and application to scenarios of future climate change and human engineering are hampered by a lack of data.

The tool presented in this study can provide typical abiotic factors that are input for habitat suitability models within minutes. Therefore, the output can lead to a first-order estimation of the potential habitat (Figure 13) based on species specific preferences (Table 1). The potential habitat area based on tool output generally overpredicts the measured habitat. First, the tool always predicts a zone with intertidal area because the predicted hypsometry is stretched between the high water line and maximum channel depth. Second, other constraining biotic and abiotic factors were excluded, such as the competition between species and the water quality. Lastly, the settling and growth conditions may deviate from the conditions under which species occur in a later life stage. For example, seedlings of salt marsh species generally only settle when peak flow is below 0.25 ms$^{-1}$ and, in their initial life stage, they may only sustain flows up to 0.5 ms$^{-1}$ [65]. Nevertheless, the application of the tool for ecological assessment is of value because it is capable of providing abiotic factors of individual systems and scenarios within minutes, rather than days to months for numerical modelling and in situ measurements.

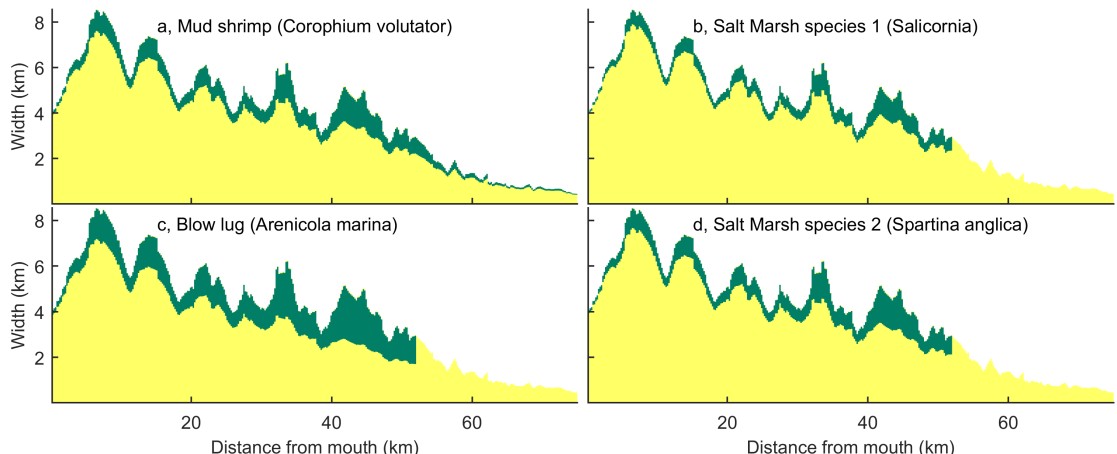

**Figure 13.** Potential habitat area (in green) as part of total width for the species in Table 1.

**Table 1.** Habitat preferences of species used in this study based on MarLIN database.

| Name | Depth Range (-) | Flow Velocity (ms$^{-1}$) | Salinity (ppt) |
|---|---|---|---|
| Mud shrimp (*Corophium volutator*) | Intertidal | <0.5 | 2–40 |
| Blow lug (*Arenicola marina*) | Intertidal | <4.0 | 18–40 |
| Salt marsh species 1 (*Salicornia*) | Intertidal above Mean Sea Level | <0.5 | 18–40 |
| Salt marsh species 2 (*Spartina anglica*) | Intertidal above −0.5*a* | <1.0 | 18–40 |

4.2.2. Application for Management of Estuaries

The main channels in estuaries are often used as shipping fairways, while the intertidal area forms important ecological habitat [2]. Given a certain channel width and depth required for shipping, the empirical predictions of bed elevations that result from this study can estimate volumes that need to be dredged and typical locations where this is necessary. While, for many of the present-day

shipping fairways, bathymetry is available, this application may be relevant for quick assessments by consultancy in developing countries with growing economies for the construction of fairways, harbours or studies on land loss in these areas [66,67].

Another implication from this study is that the wider an estuary is relative to the fitted converging shape, the wider the zone of intertidal area will be. Bed elevations and basin hypsometry form an indication of possible habitat composition [35] and low-dynamical intertidal areas are generally labelled as valuable areas [68] because they are suitable for settling and feeding. The output can indicate which areas satisfy these conditions (e.g., suitable flow conditions) and which areas can be transformed into habitat by disposal of dredged sediment. For example, in the Western Scheldt, dredged sediment has been disposed on the edges of tidal flats to maintain intertidal area [68], which aids in achieving *European Union Natura 2000* regulations [69]. This approach requires bathymetry, which may not be available at locations where new fairways or harbours are planned. The predictive tool can provide a first-order estimation in these cases.

### 4.2.3. Application in Palaeogeographic Reconstruction

Our empirical method is capable of making quantitative estimates of the bar dimensions, the number of bars and channels (Braiding Index), channel dimensions, flow conditions and salinities. The tool is applicable for palaeogeographical reconstructions [70,71] where the shape of tidal channel belts or estuaries has been identified, but wherein bar configuration is typically not recognisable and typical flow conditions are unknown. If the along-channel width profile is known and the tidal range can be estimated, the tool presented in this study can be applied. This application may be particularly relevant for high-stand estuaries that are filled with sediment. Besides the forward approach of palaeogeographical reconstructions based on estuary outline, the tool and equations also allow a reverse approach. A single measurement or preferably a few measurements of bar or channel dimensions [12,72–76] would be enough to reconstruct typical channel width and local tidal prism and therefore also typical flow conditions.

Additionally, the combination with ecological habitat analyses may improve the understanding of the evolution of estuaries. It is known that vegetation succession stabilises tidal sand bars [65,77–79], but it remains an open question to what extent this succession confines the width of estuaries on the longer term. Additionally, it is unknown what the equilibrium shape of an estuary will be under a given set of stabilising and destabilising eco-engineering species.

### 5. Conclusions

A comprehensive set of empirical relations allows estimating bathymetry, flow velocity, salinity and inundation durations for estuaries for which limited data are available. Morphological characteristics are based on the concept that the degree to which an estuary deviates from a maximum fitting converging shape is reflected in the locations where intertidal bars are found, which also determines the bed elevation distributions. This study provides a new correlation between depth below high water level and peak tidal flow velocity, based on numerical model outcomes, which complements the bed elevation predictions and allows for salinity predictions. The resulting predictions are within a factor two of the measured or modelled values, but generally better. Largest deviations between predicted and modelled or measured occur for flow velocities when they peak above 2 ms$^{-1}$, for salinity at the transition from brackish to fresh water and for bed elevations when the tidal range is low for the depth at the mouth. The end-member cases of river-dominated delta branches and tide-dominated delta branches, i.e., rivers and ideal estuaries, are most sensitive to depth at both boundaries. In these cases, when possible, it should be measured in a single cross-section and specified as input to prevent misprediction.

We conclude that the tool is suitable for alluvial estuaries filled with tidal bars, for relatively wide valleys that partially filled with bars and for tidal basins. The results are usable to quantify

valuable ecological habitat area, to estimate dredging volumes in management of estuaries or to make palaeographic reconstructions when limited data are available.

**Supplementary Materials:** The following is available at http://www.mdpi.com/2072-4292/10/12/1915/s1, A zip file with tool code and instructions to use the tool. The ZIP contains: (1) 'Instructions.pdf' with instructions on how to use the tool; (2) 'Input_variables.xls', which is file where input parameters need to be specified; (3) '.csv' files with examples of along-channel width profiles; (4) 'Model_v1_0.py', which is the tool code that needs to be run in Python; (5) 'Examples' folder with examples of the output for the Western Scheldt.

**Author Contributions:** Contributions were in following proportions to conception and design, tool programming, data collection and analysis, conclusions and manuscript preparation: J.R.F.W.L. (65%, 50%, 60%, 85%), S.L.V. (5%, 50%, 20%, 5%), W.M.v.D. (5%, 0%, 10%, 5%), S.S. (5%, 0%, 5%, 0%), and M.G.K. (20%, 0%, 5%, 5%).

**Funding:** J.R.F.W.L., W.M.v.D. and M.G.K. were supported by the Dutch Technology Foundation TTW (grant Vici 016.140.316/13710 to M.G.K., which is part of the Netherlands Organisation for Scientific Research (NWO), and is partly funded by the Ministry of Economic Affairs). S.L.V. was supported by Future Deltas, Utrecht University (seed grant to J.R.F.W.L.). S.S. was supported by an ERC Consolidator grant (647570) to M.G.K.

**Acknowledgments:** This work is part of the PhD research of J.R.F.W.L. and MSc research of S.L.V. We thank three anonymous reviewers for their comments on the manuscript. We thank Rijkswaterstaat for providing bathymetry of the Western Scheldt and Edwin Elias for providing the model of the Columbia River. Discussion with Sepehr Eslami Arab, Barend van Maanen and Tjeerd Bouma helped to improve the manuscript. The code of the tool is made available open access online with the manuscript and through GitHub (https://github.com/JasperLeuven/EstuarineMorphologyEstimator). Other data sources have been referenced in the text.

**Conflicts of Interest:** The authors declare no conflict of interest.

## Appendix A. Supplementary Figures and Table

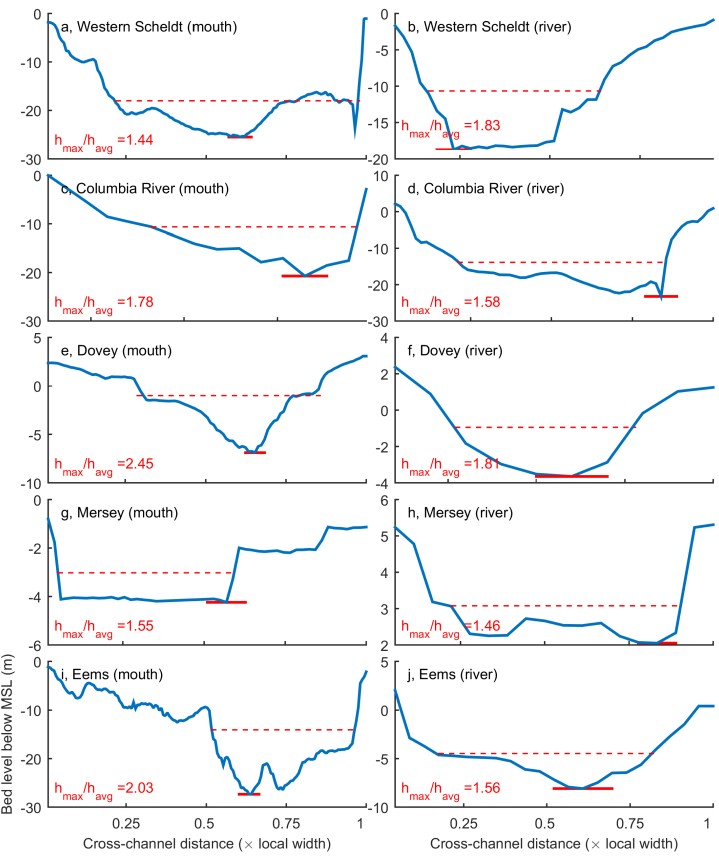

**Figure A1.** Cross-sectional profiles at the mouth (**left**) and landward river boundary (**right**) for five estuaries. The maximum depth (solid red line) and an approximation of the average water depth (dashed line) are indicated. The ratio between maximum depth and average depth, which is used in Equations (12) and (8), is given in the figures. Average ratio for the geometry of the estuary mouth is 1.85 and for the landward river boundary 1.65.

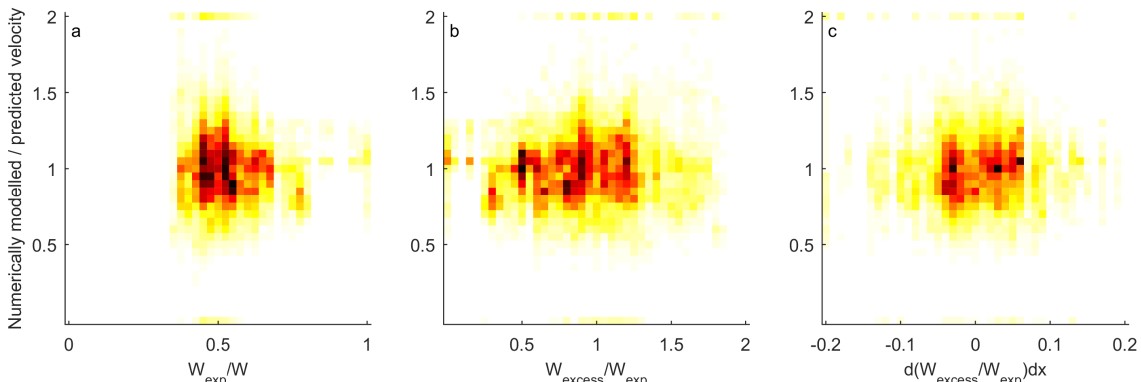

**Figure A2.** Measured velocity from the model divided by predicted velocity from the tool against three expressions of excess width. Lack of trend between velocity deviation and width measures indicates that velocity predictions will not improve from including width in the regression. (**a**) ideal width divided by local width, (**b**) excess width divided by ideal width and (**c**) along-channel derivative of excess width divided by ideal width. Colours in the plot become more yellow with increasing point density.

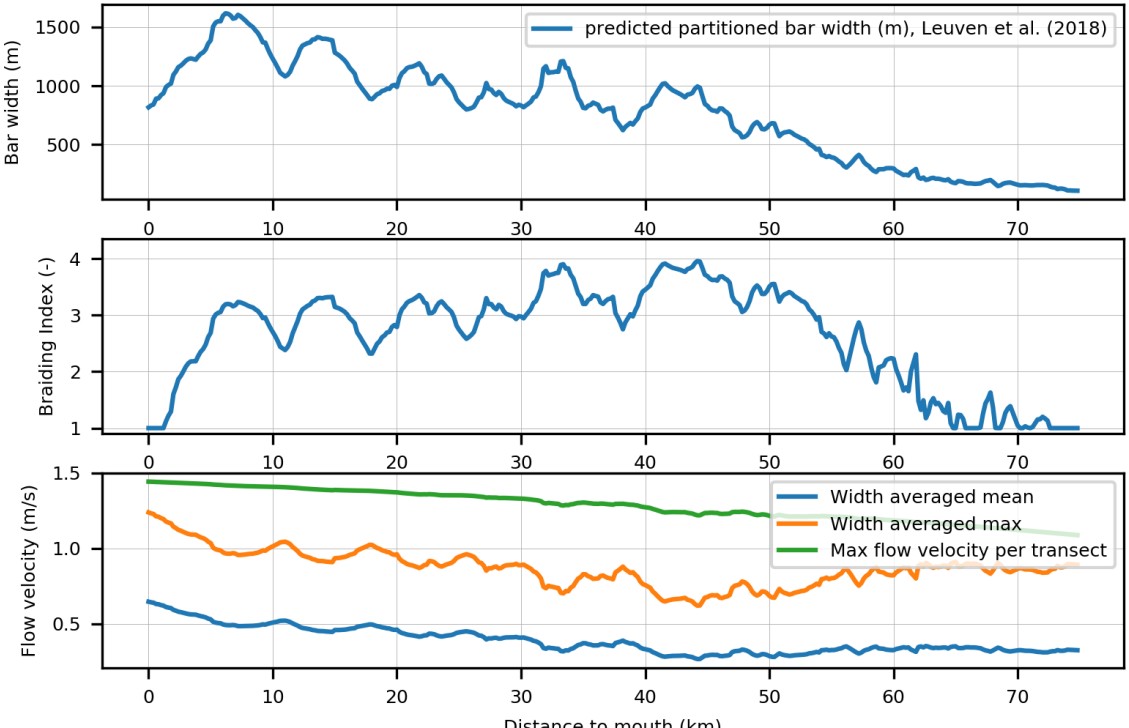

**Figure A3.** Tool output for the Western Scheldt. (**top**) predicted along-channel bar width, (**middle**) predicted along-channel braiding index and (**bottom**) the cross-sectional maximum and average of the predicted along-channel peak tidal flow velocity and the cross-sectional average of the mean flow velocity over half a tidal cycle.

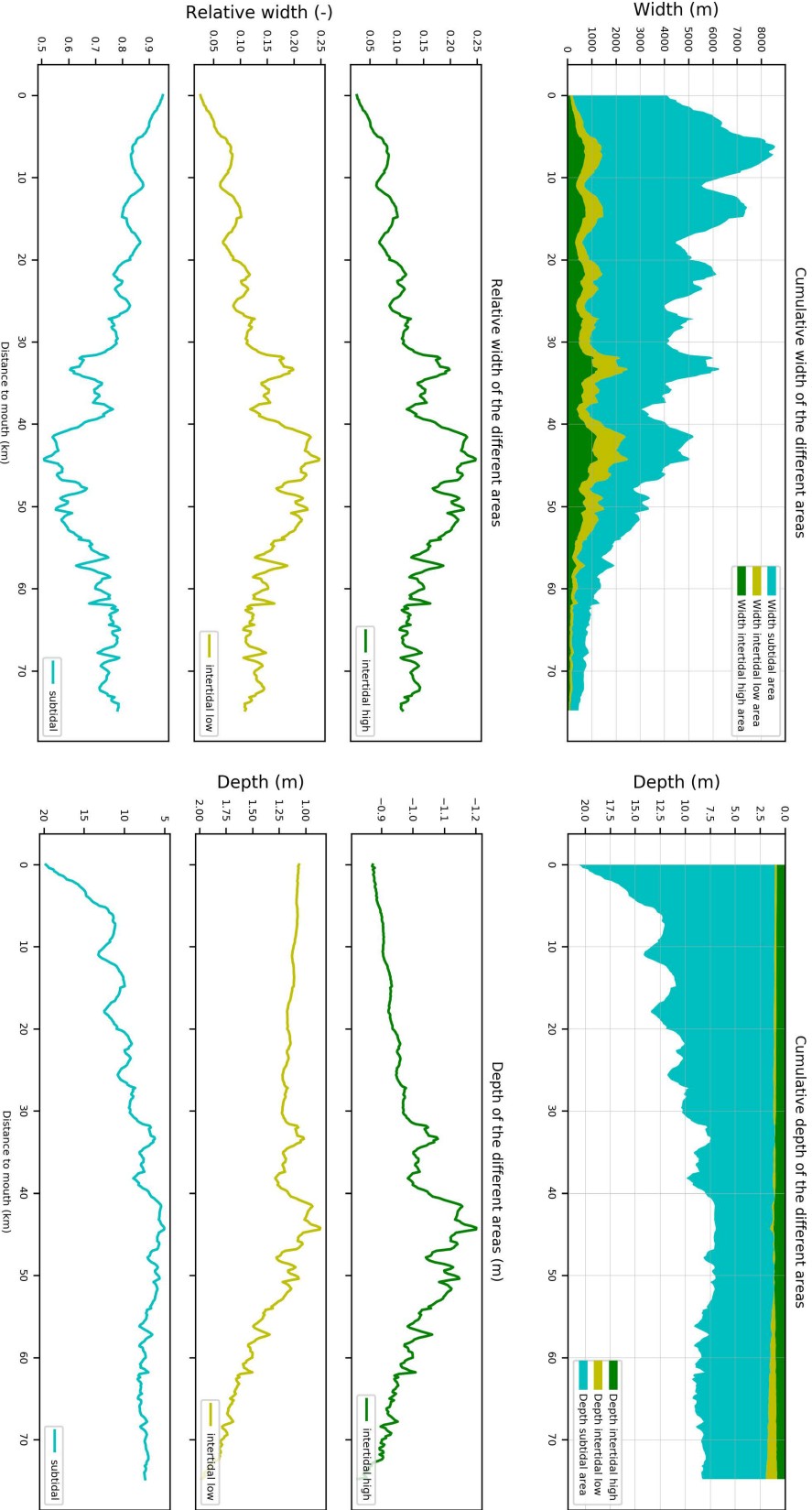

**Figure A4.** *Cont.*

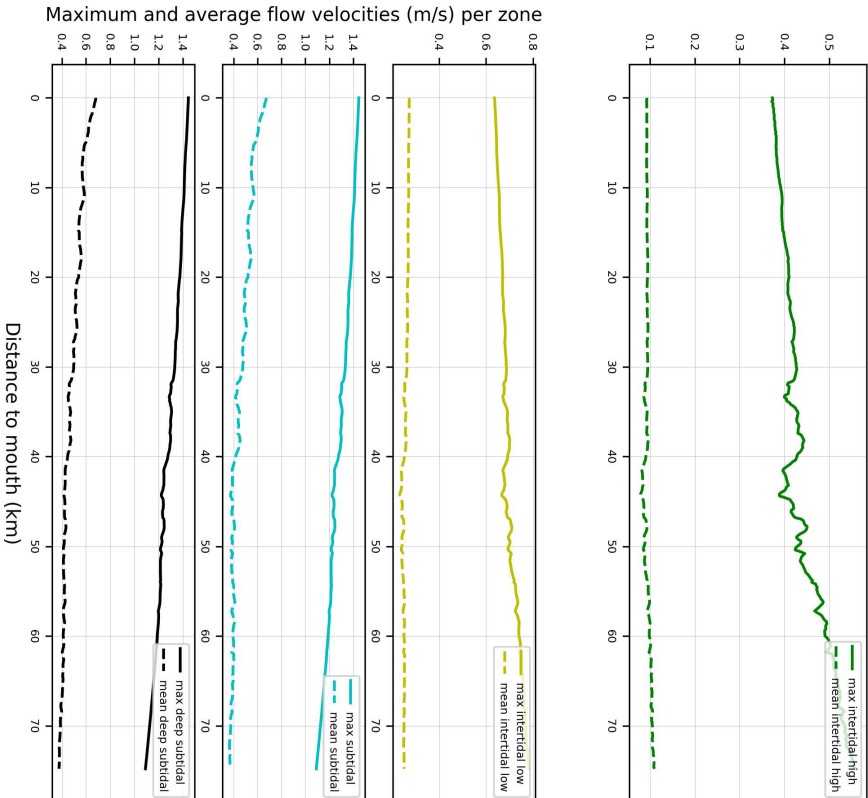

**Figure A4.** Tool output for the Western Scheldt. (**Top**) along-channel width of subtidal, intertidal low and intertial high zones; (**center**) average depth of subtidal, intertidal low and interial high zones; (**bottom**) average and maximum flow velocity of subtidal, intertidal low and interial high zones.

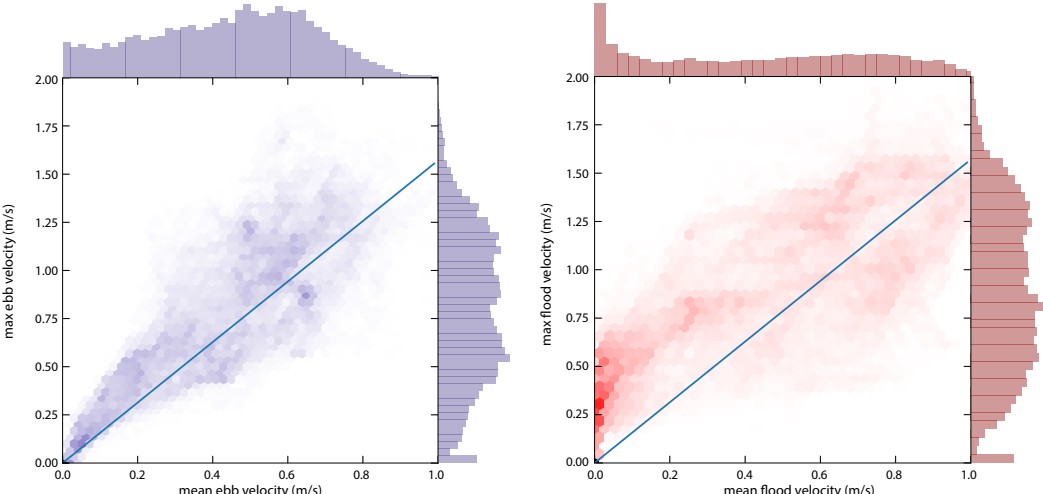

**Figure A5.** (**Left**) maximum ebb velocity as a function of mean ebb velocity; (**Right**) maximum ebb velocity as a function of mean ebb velocity. Histograms indicate occurrence within bins and colours in the plot become darker with increasing density. Drawn lines indicate the relation $x = 2/\pi y$, which is the expected average velocity when calculated as the average of half a sine function with an amplitude equal to the peak velocity.

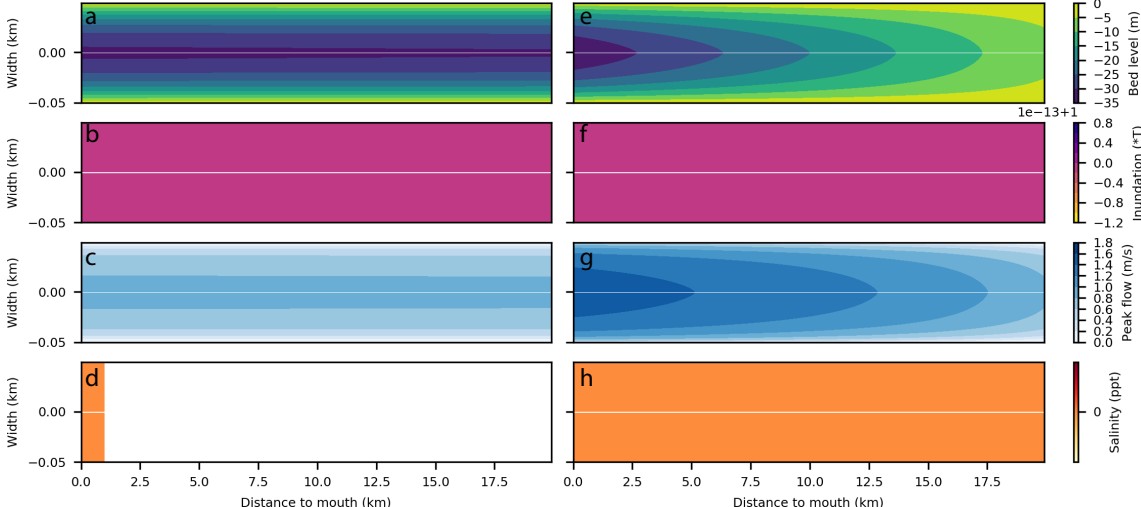

**Figure A6.** Resulting predictions of bed elevation with respect to the high water level, inundation duration, peak flow velocity and salinity for a river based on dimensions of a distributary of the Saskatchewan River, Cumberland Lake (Saskatchewan (Canada), coordinates: 54°04′N 102°22′W) (Figure 2a). In the left panels (**a**–**d**) the boundary condition for maximum depth at the mouth and the landward end are both based on hydraulic geometry for rivers (Equations (6)–(8)). In the right panels (**e**–**h**), the boundary condition for maximum depth at the mouth is based on the tidal prism relation (Equations (10) and (12)) and boundary condition at the landward end is based on hydraulic geometry for rivers (Equations (8)–(6)). See Bolla Pittaluga et al. [80] for measurements in this river.

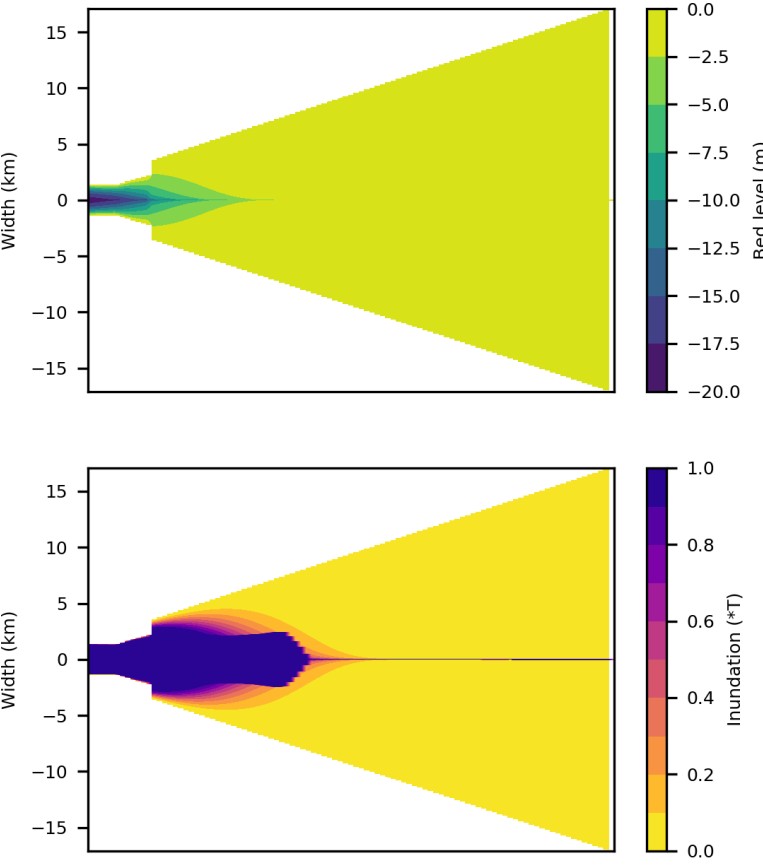

**Figure A7.** *Cont.*

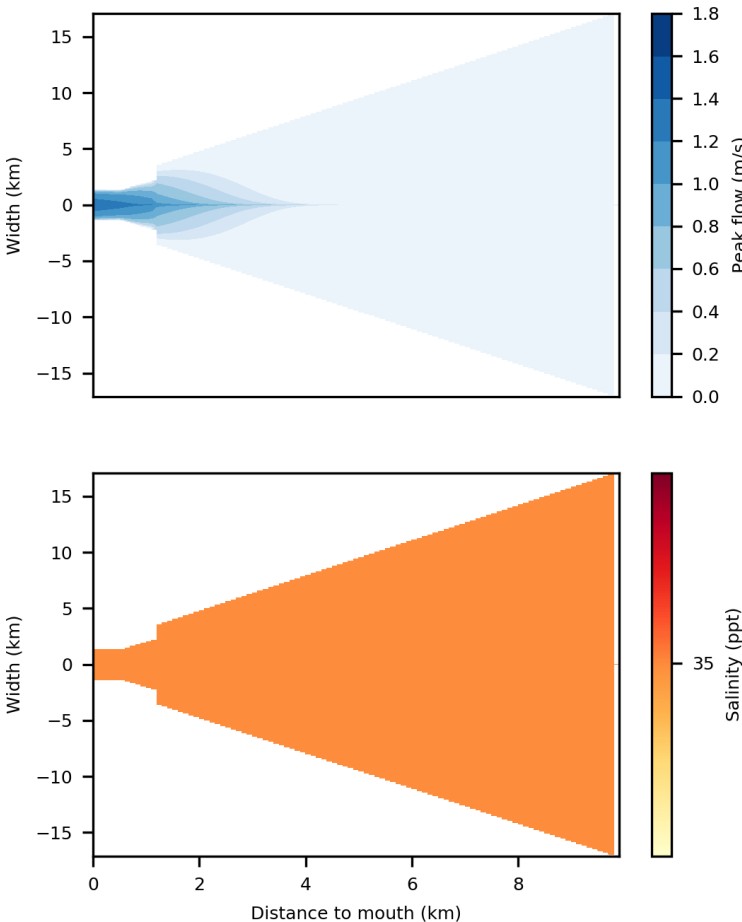

**Figure A7.** Resulting predictions of bed elevation with respect to the high water level, inundation duration, peak flow velocity and salinity for a tidal basin with dimensions based on the Ameland inlet.

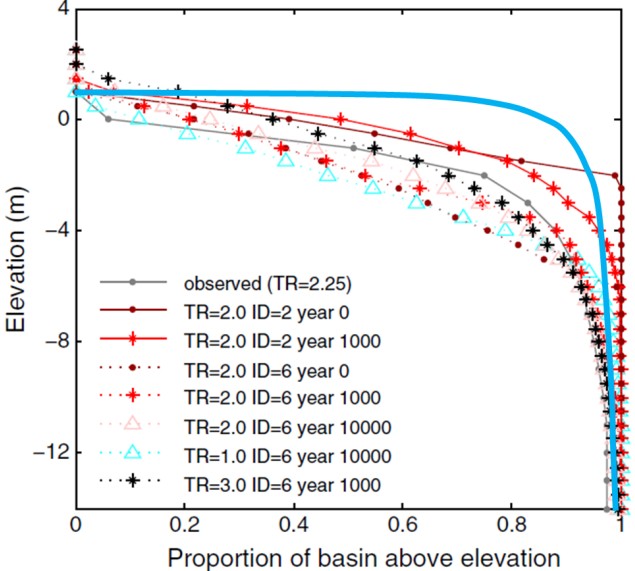

**Figure A8.** The blue solid line indicates the resulting basin hypsometry from the prediction for a tidal basin with dimensions based on the Ameland inlet (Figure A7) plotted in Figure 7 of van Maanen et al. [31].

**Table A1.** List of symbols used.

| Symbol | Units | Variable |
|---|---|---|
| $A_{csa,m}$ | [m$^2$] | Cross-sectional area at the mouth |
| $A_{csa}(x)$ | [m$^2$] | Cross-sectional area at coordinate x |
| $a(m)$ | [m] | Tidal range at the mouth |
| $a(x)$ | [m] | Tidal range at the mouth |
| $BI(x)$ | [-] | Braiding Index at coordinate x |
| $C_m$ | [m$^{0.5}$s$^{-1}$] | Chezy roughness at the estuary mouth |
| $D_m$ | [m$^2$s$^{-1}$] | Dispersion coefficient at the mouth of the estuary |
| $D(x)$ | [m$^2$s$^{-1}$] | Dispersion coefficient at coordinate x |
| $\Delta\rho$ | [-] | Density difference between salt and fresh water |
| $E$ | [m] | Tidal excursion length |
| $g$ | [ms$^{-2}$] | Gravitational acceleration |
| $h_{max,r}$ | [m] | Maximum channel depth at the landward boundary |
| $\overline{h_r}$ | [m] | Average channel depth at the landward boundary |
| $h_{max,m}$ | [m] | Maximum channel depth at the estuary mouth |
| $\overline{h_m}$ | [m] | Average channel depth at the estuary mouth |
| $h_z(x,y)$ | [-] | Dimensionless bed elevation at coordinate (x,y) |
| $h(x,y)$ | [m] | Dimensional bed elevation at coordinate (x,y) |
| $I(x,y)$ | [× t] | Inundation duration |
| $K$ | [m$^2$s$^{-1}$] | Empirical equation for the longitudinal mixing coefficient |
| $K_s$ | [-] | Longitudinal Van der Burgh mixing coefficient in Savenije [21] |
| $K_g$ | [-] | Longitudinal Van der Burgh mixing coefficient in Gisen et al. [23] |
| $L_W$ | [m] | Width convergence length |
| $L_{A_{csa}}$ | [m] | Cross-sectional area convergence length |
| $N_r$ | [-] | Estuarine Richardson number |
| $P(x)$ | [m$^3$] | Local tidal prism at coordinate x |
| $P(m)$ | [m$^3$] | Local tidal prism at the estuary mouth |
| $Q_b$ | [m$^3$s$^{-1}$] | Bankfull river discharge |
| $Q_r$ | [m$^3$s$^{-1}$] | Average river discharge |
| $r$ | [-] | Coefficient in the Strahler [33] equation |
| $r_s$ | [-] | Storage ratio (intertidal area/subtidal area) |
| $\rho_w$ | [kgm$^{-3}$] | Water density |
| $S_m$ | [ppt] | Salinity at the estuary mouth |
| $S_r$ | [ppt] | Salinity at the landward boundary |
| $S_b(x)$ | [ppt] | Salinity at coordinate *x* [22] |
| $S_g(x)$ | [ppt] | Salinity at coordinate *x* [23] |
| $S_s(x)$ | [ppt] | Salinity at coordinate *x* [21] |
| $s$ | [s] | Distance between the mouth and landward boundary |
| $s_m$ | [-] | Shape factor of the channel at the estuary mouth |
| $s_r$ | [-] | Shape factor of the channel at the landward boundary |
| $t$ | [s] | Duration of half a tidal cycle |
| $u_{avg}(x,y)$ | [ms$^{-1}$] | Tidal average flow velocity at coordinate (x,y) |
| $u_{peak}(x,y)$ | [ms$^{-1}$] | Peak tidal flow velocity at coordinate (x,y) |
| $u_{peak}(mouth,y)$ | [ms$^{-1}$] | Peak tidal flow velocity at the estuary mouth |
| $W_{excess}(x)$ | [m] | Excess width at coordinate *x* |
| $W_{ideal}(x)$ | [m] | Ideal width at coordinate *x* |
| $W_m$ | [m] | Width at the estuary mouth |
| $W_r$ | [m] | Width at the landward boundary |
| $W(x)$ | [m] | Local width at coordinate *x* |
| $w_{bar}(x)$ | [m] | Predicted bar width at coordinate *x* |
| $w_{barssummed}(x)$ | [m] | Summed width of bars at coordinate *x* |
| $x$ | [m] | Streamwise coordinate measured from the mouth along the centreline |
| $y$ | [m] | Coordinate perpendicular to the centreline |
| $z$ | [-] | Coefficient in the Strahler [33] equation |
| $z(x)$ | [-] | Value of *z*-coefficient at coordinate *x* |

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
