# Peer review of "Empirical Assessment Tool for Bathymetry, Flow Velocity and Salinity in Estuaries Based on Tidal Amplitude and Remotely-Sensed Imagery"

_remotesensing, doi:10.3390/rs10121915_

Reviewer 1 Report

Revision of the manuscript n. 395077.

The manuscript entitled “Empirical assessment tool for bed levels, flow velocity and salinity in estuaries based on tidal amplitude and space-borne imagery” by Leuven et al. has been revised.

The paper provides a quick tool estimation of hydrogeomorphological parameters in estuarine areas by means of a numerical model. Such parameters are useful to quantify the evaluation of ecological habitat areas and to compute the dredging volume by human interventions. The manuscript is well written, on the whole, and the aims of the paper are clearly exposed in the Introduction section. In this section I only suggest to contract the text that is too long, in my opinion, and it causes the reader to lost the aims of the work. Moreover, I suggest more clear comments on the previous papers you cited on the argument. Methods section is very detailed and the use of equations is well described. Considering that in the Method section the following subsections 2.3. (Inundation duration prediction), 2.4. (Bar pattern prediction: bar width and braiding index), and 2.5. (Flow velocity prediction) are very short, could be useful to join them in one more wide subsection, if it is possible. Results are clearly illustrated and only at Lines 253-260 I don’t understand the meaning of the sentence. It seems to be a circular argumentation and could be better written. Discussion is well reasoned and figures are well expounded. Conclusion are sufficient. The text titles in figures 5, 6, 7, and 11 are very small and I suggest to enlarge them. Finally, I suggest a light revision of the text by a native English speaker.

Considering the whole text I think that the manuscript is suitable for publication on the Remote Sensing Journal, after minor revisions.

Author Response

The manuscript entitled “Empirical assessment tool for bed levels, flow velocity and salinity in estuaries based on tidal amplitude and space-borne imagery” by Leuven et al. has been revised. The paper provides a quick tool estimation of hydrogeomorphological parameters in estuarine areas by means of a numerical model. Such parameters are useful to quantify the evaluation of ecological habitat areas and to compute the dredging volume by human interventions.

Thank you for your review.

The manuscript is well written, on the whole, and the aims of the paper are clearly exposed in the Introduction section. In this section I only suggest to contract the text that is too long, in my opinion, and it causes the reader to lost the aims of the work. Moreover, I suggest more clear comments on the previous papers you cited on the argument.

We revised the introduction and removed sentences where possible and added comments on previous papers where necessary.

Methods section is very detailed and the use of equations is well described. Considering that in the Method section the following subsections 2.3. (Inundation duration prediction), 2.4. (Bar pattern prediction: bar width and braiding index), and 2.5. (Flow velocity prediction) are very short, could be useful to join them in one more wide subsection, if it is possible.

We decided to leave it as it is, because now the sections directly correspond to the individual output maps as shown in Fig. 3 and later figures with the results (e.g. Figs. 5-6), which allows the reader to quickly locate the relevant information for each output map.

Results are clearly illustrated and only at Lines 253-260 I don’t understand the meaning of the sentence. It seems to be a circular argumentation and could be better written.

We rephrased this paragraph as: “Before, we had to estimate a typical peak flow velocity, because it was required for the estimation of the tidal excursion length (Eq. 9). At that point, a value of 1 m/s was assumed (Savenije, 2006). This assumption only partly influences the output flow velocity, which depends on bed elevation, which in turn is controlled by the estuary planform shape, the tidal range and the tidal excursion length. Nevertheless, output maps justify the assumption of 1 m/s for the purpose of calculating the tidal excursion length. The average peak tidal flow velocity is 1 m/s for the Western Scheldt and 1.2 m/s for the Columbia River when averaged over the surface area of a typical tidal excursion length (Figs. 5-6). Median values of peak tidal flow velocity per transect are also remarkably close to the typical value proposed by Savenije (2006) along the entire estuary (Fig. 7 ac).”

Discussion is well reasoned and figures are well expounded. Conclusion are sufficient. The text titles in figures 5, 6, 7, and 11 are very small and I suggest to enlarge them. Finally, I suggest a light revision of the text by a native English speaker.

Thank you for pointing out. We enlarge the figures 5, 6, 7 and 11 and revised the text on English language, including details comments from Reviewer 2.

Considering the whole text I think that the manuscript is suitable for publication on the Remote Sensing Journal, after minor revisions.

Reviewer 2 Report

This paper develops and applies a Python tool for estimating bed levels, flow velocity, and salinity in estuaries based on remotely-sensed imagery and tidal amplitude. This tool is a valuable contribution, because there are many geological or ecological applications in which data on estuarine bathymetry, flow velocity, and salinity are needed, but these data can be difficult to obtain. Estimating these parameters using only remotely-sensed imagery and tidal amplitude will therefore be very useful. One mild criticism I have is that the remote-sensing component of the paper is relatively minor, being used just to generate the along-channel width profile. But I still think the paper is a good fit for Remote Sensing because it does a nice job of integrating remote-sensing with other data sources in order to create a useful product. The approach taken is reasonable, and the paper is mostly well-written and organized. My main criticism of the paper is the use of only one hydraulic geometry equation, with no discussion of the uncertainties inherent in hydraulic geometry. I also have some minor suggested revisions, which I think will help to improve the clarity of the presentation. See below for specific comments.

Line 8: The term “bed levels” is used throughout the paper where I would use “bathymetry” (or possibly “bed elevation”), but maybe this is a difference between the fluvial community (of which I am part) versus the estuarine research community.

Line 36: Another terminological quibble: here aerial photography (implying airborne) is mentioned as the data source, whereas the title of the paper mentions space-borne (i.e. satellite) imagery. I’m sure this approach could be used with either satellite data (assuming the estuary is large enough to be resolved) or airborne imagery, where available. So maybe keep it general by just saying “remotely-sensed imagery”.

Figure 2: Some inset location maps would be helpful.

Line 92: One thing that is not clear to me is how the width-channel profile is obtained. It looks like it is an input to the tool, so I assume the user must manually calculate it based on the imagery beforehand and that the Python script doesn’t do anything with the actual imagery. Some clarity here would be appreciated.

Lines 124-126: How is the exact location of the estuary mouth and the landward boundary of the estuary determined, given that rivers gradually transform into the estuarine environment?

Lines 126-158: Hydraulic geometry is fraught with uncertainty, mainly because it is an empirical construct with little to no theoretical support. In the absence of observed data on river discharge, it is often the best estimate that can be made. But given their empirical nature, hydraulic geometry equations are highly sensitive to the rivers for which they are developed, and regionally specific equations are often derived. The Hey and Thorne equation used here is based on gravel-bed rivers in the UK and is unlikely to produce reliable estimates of bankfull discharge when applied to other river types or regions. So, it might be helpful to include in the tool some other regional hydraulic geometry equations so that users can choose the one most appropriate for their region.

Figure 4: I would like to see a legend that shows the magnitudes associated with the colors.

Line 170: Not sure why it says the summed width of bars “approaches” the excess width when Equation 15 says the summed width of bars is equal to the excess width?

Line 180: It would be good to see some justification for believing that the relationship between bed elevation and typical flow velocity is valid not only for the Western Scheldt, but for other estuaries as well.

Line 201: I assume that since the hydraulic geometry equations predict bankfull discharge, the salinity calculations are based on bankfull discharge as well?

Lines 211-214: In the validation, was observed river discharge used, or was it based on the hydraulic geometry?

Line 228: You don’t need to provide the geographic coordinates in the text, as those are provided in the maps.

Lines 239-242: This section seems redundant after the previous explanation of the validation and test cases.

Figures 5 and 6: The caption for the velocity plot reads “Peak flow”, but it should be “Peak flow velocity”.

Line 247: Typo: “levers” should be “levels”.

Line 270: “a factor 2” should be “a factor of 2”.

Figure 7: The x-axis labels for parts b and d are a bit confusing. I would make them the same as for parts a and c.

Line 335: It looks like there’s a reference missing.

Figure 11: “Peak flow” should be “Peak flow velocity”.

Lines 383-385: This section seems unnecessary.

Line 396: “Sensitivity tool output” should be “Sensitivity of tool output”.

Table 1, Figure 13: Since “Salt Marsh” is not a species, you could rename these items “Salt marsh species 1 (Salicornia) and “Salt marsh species 2 (Spartina anglica)”.

Line 462: “growing economy” should be “growing economies”.

Line 478: “to which extent” should be “to what extent”.

Line 485: I’m confused by this sentence, but I think it makes more sense if “reflects in” is changed to “is reflected in”.

Author Response

This paper develops and applies a Python tool for estimating bed levels, flow velocity, and salinity in estuaries based on remotely-sensed imagery and tidal amplitude. This tool is a valuable contribution, because there are many geological or ecological applications in which data on estuarine bathymetry, flow velocity, and salinity are needed, but these data can be difficult to obtain. Estimating these parameters using only remotely-sensed imagery and tidal amplitude will therefore be very useful.

Thank you for your review and detailed comments. Please find our point-by-point response below in blue.

One mild criticism I have is that the remote-sensing component of the paper is relatively minor, being used just to generate the along-channel width profile. But I still think the paper is a good fit for Remote Sensing because it does a nice job of integrating remote-sensing with other data sources in order to create a useful product. The approach taken is reasonable, and the paper is mostly well-written and organized.

Even though this article definitely has a basis in remote sensing, we anticipated this, which is why we asked the editorial board whether our paper would fit the scope of the Special Issue "Remote Sensing of Flow Velocity, Channel Bathymetry, and River Discharge" before submission. They encouraged our submission.

While the component of Remote Sensing is relatively minor, we explicitly addresses flow velocity and channel depth. We feel this paper provides an excellent opportunity to obtain these characteristics when only remotely sensed imagery is available, which is now openly available and easy to use for the world with Google Earth and Python. Moreover, it may be combined later with other Remote Sensing techniques to optimise the results (e.g. predict the location of individual channels and bars) and make it more widely applicable (e.g. delta branches, ebb-tidal deltas, etc.).

My main criticism of the paper is the use of only one hydraulic geometry equation, with no discussion of the uncertainties inherent in hydraulic geometry.

We now acknowledge and discuss the limitations and uncertainties of HG in the revised manuscript, also following the recommendation of the Academic Editor. Moreover, we emphasize that any manual input overrides the prediction and that thus any user can choose/use their own predictor (lines 110-111). We added the following paragraph to the text (lines 169-176):

“Hydraulic geometry is an empirical construct and should be used with care. However, it is often the best estimate in absence of observed data. The resulting predictions are dependent on the specific region or river type for which they were developed.  Here we implemented, and used, the Hey and Thorne (1986) equation as an example. Nevertheless, users of the tool can specify a measured channel depth and river discharge as input when available, because it overrides the calculation with Eqs. 6-7 and therefore implicitly impose a hydraulic geometry. Alternatively, the user can calculate these values with a hydraulic geometry equation suitable for the type of river considered or implement their own equation in the source code.”

Minor comments

Line 8: The term “bed levels” is used throughout the paper where I would use “bathymetry” (or possibly “bed elevation”), but maybe this is a difference between the fluvial community (of which I am part) versus the estuarine research community.

We replaced ‘bed levels’ either with ‘bathymetry’ or ‘bed elevation’ throughout the manuscript.

Line 36: Another terminological quibble: here aerial photography (implying airborne) is mentioned as the data source, whereas the title of the paper mentions space-borne (i.e. satellite) imagery. I’m sure this approach could be used with either satellite data (assuming the estuary is large enough to be resolved) or airborne imagery, where available. So maybe keep it general by just saying “remotely-sensed imagery”.

Replaced.

Figure 2: Some inset location maps would be helpful.

We tried this, but the figure is too small to include 5 insert maps. Given that the coordinates are in the figure and the countries are listed in the caption, we feel that the reader has sufficient information to located the systems.

Line 92: One thing that is not clear to me is how the width-channel profile is obtained. It looks like it is an input to the tool, so I assume the user must manually calculate it based on the imagery beforehand and that the Python script doesn’t do anything with the actual imagery. Some clarity here would be appreciated.

Thank you for pointing this out. We now added a paragraph at the start of section 2.2 outlining how one can obtain the channel width profile (lines …-…):

“From remotely-sensed imagery, the first step is to obtain the along-channel width profile. Multiple options are available ranging from manually measuring the width at equally spaced transects to software tools that calculate river widths from remotely-sensed imagery (e.g. Pavelsky & Smith, 2008; Donchyts et al., 2016). Here we used the following approach (following Leuven et al., 2018): first we digitised the estuary planform in GIS software. Second, the polygon was imported in GIS software and the channel centreline was automatically extracted (e.g. Polygon To Centerline, ArcGIS function). Last, transects were drawn equally spaced on the centreline (e.g. Generate Transects Along Lines, ArcGIS function) and cropped with the planform polygon. The length of successive transects gave the along-channel width profile.”

Lines 124-126: How is the exact location of the estuary mouth and the landward boundary of the estuary determined, given that rivers gradually transform into the estuarine environment?

We added here that “We followed the same guidelines on the selection of the landward and seaward boundary as in Leuven et al. (2018), given on p. 766.”

Lines 126-158: Hydraulic geometry is fraught with uncertainty, mainly because it is an empirical construct with little to no theoretical support. In the absence of observed data on river discharge, it is often the best estimate that can be made. But given their empirical nature, hydraulic geometry equations are highly sensitive to the rivers for which they are developed, and regionally specific equations are often derived. The Hey and Thorne equation used here is based on gravel-bed rivers in the UK and is unlikely to produce reliable estimates of bankfull discharge when applied to other river types or regions. So, it might be helpful to include in the tool some other regional hydraulic geometry equations so that users can choose the one most appropriate for their region.

See our reply to the reviewer’s comment on this above.

Figure 4: I would like to see a legend that shows the magnitudes associated with the colors.

Added.

Line 170: Not sure why it says the summed width of bars “approaches” the excess width when Equation 15 says the summed width of bars is equal to the excess width?

Changed.

Line 180: It would be good to see some justification for believing that the relationship between bed elevation and typical flow velocity is valid not only for the Western Scheldt, but for other estuaries as well.

At this point, we do not have sufficient hydronamic data to assess the validity for other estuaries, which is why we urge the need for more flow velocity data in lines 356-359. In the results, we assess the validity for the Columbia River and other end member systems. Moreover, a comparison with the tidal prism based relation and numerically modelled data (Fig. 9) shows that the spatial pattern is better represented by the depth relation than based on the tidal prism.

Line 201: I assume that since the hydraulic geometry equations predict bankfull discharge, the salinity calculations are based on bankfull discharge as well?

This is the average river discharge. We added ‘average’ here. Furthermore, the difference in symbology between bankfull discharge and average discharge is also shown in supplementary table A1.

Lines 211-214: In the validation, was observed river discharge used, or was it based on the hydraulic geometry?

We added in the text: “For validation, channel depth at the landward boundary was calculated in the tool with measured river discharge.”

Line 228: You don’t need to provide the geographic coordinates in the text, as those are provided in the maps.

Removed.

Lines 239-242: This section seems redundant after the previous explanation of the validation and test cases.

Removed.

Figures 5 and 6: The caption for the velocity plot reads “Peak flow”, but it should be “Peak flow velocity”.

Changed.

Line 247: Typo: “levers” should be “levels”. Line 270: “a factor 2” should be “a factor of 2”.

Changed.

Figure 7: The x-axis labels for parts b and d are a bit confusing. I would make them the same as for parts a and c.

Done.

Line 335: It looks like there’s a reference missing.

This was a part we forgot to remove.

Figure 11: “Peak flow” should be “Peak flow velocity”.

Changed.

Lines 383-385: This section seems unnecessary.

Removed.

Line 396: “Sensitivity tool output” should be “Sensitivity of tool output”.

Changed.

Table 1, Figure 13: Since “Salt Marsh” is not a species, you could rename these items “Salt marsh species 1 (Salicornia) and “Salt marsh species 2 (Spartina anglica)”.

Changed.

Line 462: “growing economy” should be “growing economies”.

Changed.

Line 478: “to which extent” should be “to what extent”.

Changed.

Line 485: I’m confused by this sentence, but I think it makes more sense if “reflects in” is changed to “is reflected in”.

Changed.

Reviewer 3 Report

The paper is well written both in terms of organization, flow, and plausibility as well as the use of English language. It is a rare instance to be able to read a manuscript this well constructed and construed. 

I found the paper to be highly interesting as it builds on existing relationships and knowledge and the expands to new tools by further developing these relationships through including new ideas. This is perhaps to be expected considering that this is the outcome of PhD level research. 

I further much appreciate the decision to make this tool available to the general audience, a very big plus that increases the relevancy of the work. The only point of critique I have on this aspect is that the general user needs to acquire familiarity with Python before she/he can use it. A packed downloadable to expands on mouse click on a desktop with a simple GUI (which can be coded in Python also) would have been nice. Yet, this will not take away from the quality of the paper and is not a major shortfall. Just something to consider in case an updated version would come about. 

I truly like the presentation of the various estuaries and also the attempt to explore the tool's performance for set ups at the range limits of typical estuaries. This goes hand in hand with the very honest disclosure of assumptions made and also shortcomings, i.e. it does not always work perfectly. I would have liked to see an example where it fully fails, thus expanding the applicability horizon, or rather the limitations of it. Right now it seems that the full range of estuaries have been explored (the "black" and the "white" have been identified and everything else is a shade of grey), but have they? Is this really it, there are no other types, or existing types that have some special deviations to them, potentially rendering the tool less effective? Again, this is just a suggestion to the authors to include a discussion point that frames this question. Would be nice for the reader to know with more certainty. 

Overall this is a splendid paper ... with one question remaining: why was it submitted to this journal? There is no RS component presented at all even though I assume that the along channel width profile may be extracted from RS imagery. 

Author Response

The paper is well written both in terms of organization, flow, and plausibility as well as the use of English language. It is a rare instance to be able to read a manuscript this well constructed and construed. I found the paper to be highly interesting as it builds on existing relationships and knowledge and the expands to new tools by further developing these relationships through including new ideas. This is perhaps to be expected considering that this is the outcome of PhD level research. 

Thank you for your review.

I further much appreciate the decision to make this tool available to the general audience, a very big plus that increases the relevancy of the work. The only point of critique I have on this aspect is that the general user needs to acquire familiarity with Python before she/he can use it. A packed downloadable to expands on mouse click on a desktop with a simple GUI (which can be coded in Python also) would have been nice. Yet, this will not take away from the quality of the paper and is not a major shortfall. Just something to consider in case an updated version would come about. 

We now appended a .pdf with instructions on how to use the tool as Supplementary Material, such that even a user without pre-knowledge of Python is able to use the tool. We tested whether colleagues without experience in Python could use the tool, which was indeed the case. More advanced users of Python will be able to modify the source code.

Instead of a GUI, we decided to use a similar approach: we now make use of a single .xls file in which the user can specify the input variables. Additionally, the along-channel width profile needs to be added as a .csv file in the same folder. Then, the user only needs to press ‘Run’ in the Python environment. All the specific steps are outlined in the .pdf with instructions.

I truly like the presentation of the various estuaries and also the attempt to explore the tool's performance for set ups at the range limits of typical estuaries. This goes hand in hand with the very honest disclosure of assumptions made and also shortcomings, i.e. it does not always work perfectly. I would have liked to see an example where it fully fails, thus expanding the applicability horizon, or rather the limitations of it. Right now it seems that the full range of estuaries have been explored (the "black" and the "white" have been identified and everything else is a shade of grey), but have they? Is this really it, there are no other types, or existing types that have some special deviations to them, potentially rendering the tool less effective? Again, this is just a suggestion to the authors to include a discussion point that frames this question. Would be nice for the reader to know with more certainty.

What the reviewer asks for is essentially what we did, so we used this comment to clarify how we used end-member cases to study the performance of the tool in the most extreme cases. Please note that these extreme cases are thus not estuaries, as the reviewer suggests. We now implemented the following paragraph in the text:

“We tested the tool’s performance against end-member systems. The dominant factors controlling the tool output are (i) the along-channel width profile, (ii) river discharge at the landward boundary and (iii) tidal range at the seaward boundary. Given these three factors, we designed three most extreme end-member cases (Fig. 2a,b,e), which were tested in addition to two estuarine cases with variable width profile (the Western Scheldt and Columbia River) (Fig. 2c,d). The first end-member is an estuary with a perfectly converging shape perfectly converging shape from the mouth to the upstream river, i.e. negligible width variation. This case was based on the dimensions of the Tran De branch of the Mekong Delta (Vietnam). The second end-member is a river with a constant channel width, i.e. no tidal range at the seaward boundary and negligible width variation. This case was based on the dimensions of a distributary of the Saskatchewan River, near Cumberland Lake (Saskatchewan, Canada). The third end-member is a tidal basin, i.e. no river discharge and landward increasing width instead of decreasing. This case was based on dimensions of the Ameland inlet and represented by a narrow mouth, an increasing channel width in landward direction and an infinitely small river at the landward boundary. It should be noted that in the last case, along-channel width is measured perpendicular to the tidal channels, which means that the transects for an idealised tidal basin are semicircles. The predictions for the Tran De branch were compared with measured data (Nguyen, 2006; Nowacki, 2015; Xing, 2017) and the predictions for the tidal basin were compared with numerical model results (Maanen, 2011; Maanen, 2013).”

Overall this is a splendid paper ... with one question remaining: why was it submitted to this journal? There is no RS component presented at all even though I assume that the along channel width profile may be extracted from RS imagery. 

Even though this article definitely has a basis in remote sensing, we anticipated this, which is why we asked the editorial board whether our paper would fit the scope of the Special Issue "Remote Sensing of Flow Velocity, Channel Bathymetry, and River Discharge" before submission. They encouraged our submission.

While the component of Remote Sensing is relatively minor, we explicitly addresses flow velocity and channel depth. We feel this paper provides an excellent opportunity to obtain these characteristics when only remotely sensed imagery is available, which is now openly available and easy to use for the world with Google Earth and Python. Moreover, it may be combined later with other Remote Sensing techniques to optimise the results (e.g. predict the location of individual channels and bars) and make it more widely applicable (e.g. delta branches, ebb-tidal deltas, etc.).